# Terahertz electric-field-driven dynamical multiferroicity in SrTiO₃

M. Basini[1,11], M. Pancaldi[2,3,11], B. Wehinger[2,4], M. Udina[5], V. Unikandanunni[1], T. Tadano[6], M. C. Hoffmann[7], A. V. Balatsky[2,8,9,10] & S. Bonetti[1,2,10✉]

The emergence of collective order in matter is among the most fundamental and intriguing phenomena in physics. In recent years, the dynamical control and creation of novel ordered states of matter not accessible in thermodynamic equilibrium is receiving much attention[1–6]. The theoretical concept of dynamical multiferroicity has been introduced to describe the emergence of magnetization due to time-dependent electric polarization in non-ferromagnetic materials[7,8]. In simple terms, the coherent rotating motion of the ions in a crystal induces a magnetic moment along the axis of rotation. Here we provide experimental evidence of room-temperature magnetization in the archetypal paraelectric perovskite SrTiO₃ due to this mechanism. We resonantly drive the infrared-active soft phonon mode with an intense circularly polarized terahertz electric field and detect the time-resolved magneto-optical Kerr effect. A simple model, which includes two coupled nonlinear oscillators whose forces and couplings are derived with ab initio calculations using self-consistent phonon theory at a finite temperature[9], reproduces qualitatively our experimental observations. A quantitatively correct magnitude was obtained for the effect by also considering the phonon analogue of the reciprocal of the Einstein–de Haas effect, which is also called the Barnett effect, in which the total angular momentum from the phonon order is transferred to the electronic one. Our findings show a new path for the control of magnetism, for example, for ultrafast magnetic switches, by coherently controlling the lattice vibrations with light.

It is now established that noncollinear magnetic order in magnetic insulators can induce electric polarization along the axis of the spiral spin structure[10]. The fundamental microscopic mechanism is the Dzyaloshinskii–Moriya interaction, which has the form $\mathbf{S}_i \times \mathbf{S}_j$ for spins $\mathbf{S}$ at sites $i$ and $j$ and which is known to promote noncollinear magnetic order[11–13]. By including full relativistic corrections owing to the spin–orbit coupling, the Dzyaloshinskii–Moriya interaction leads to a polarization $\mathbf{P}$ of the form $\mathbf{P} \propto \mathbf{e}_{ij} \times (\mathbf{S}_i \times \mathbf{S}_j)$, where $\mathbf{e}_{ij}$ is the unit vector between sites $i$ and $j$, as shown in ref. 10. On the other hand, from symmetry considerations, the permutation of space and time $t$, and of electric and magnetic fields, a magnetization $\mathbf{M}$ of the form $\mathbf{M} \propto \mathbf{P} \times \partial_t \mathbf{P}$ is expected to appear in the presence of a time-dependent polarization[7]. Within a classical picture, the motion of ions in a closed loop induces an orbital magnetic moment, which is of the order of the nuclear magneton $\mu_N \approx 10^{-3} \mu_B$ per unit cell. Inducing such ionic motion is the main idea of the experiment presented here and illustrated in Fig. 1.

SrTiO₃ (STO) is a paraelectric diamagnetic material with a cubic perovskite structure at room temperature. It is a suitable material for probing electric-field-driven dynamics, having several zone-centre phonon modes in a frequency range accessible to modern terahertz sources[14–16].

The polar ferroelectric phonon mode can be driven circularly, owing to its threefold degeneracy in the cubic crystal structure at room temperature, and it is infrared active. This mode softens significantly in frequency from $\nu \approx 2.7$ THz at temperature $T = 300$ K to $\nu \approx 0.2$ THz at $T = 5$ K (refs. 17,18), but it does not establish long-range ferroelectric order: quantum fluctuations prevent the system from remaining in one of its electronic potential minima even at zero temperature. The mode-selective drive by a strong terahertz field allows the excitation of the ferroelectric soft mode into highly anharmonic regimes, enabling coupling to modes with a different symmetry, which have recently been probed with ultrafast X-ray diffraction[19].

However, the response of the ferroelectric mode to circularly polarized, narrowband, terahertz pulses and the potential onset of magnetization due to dynamical multiferroicity are still unexplored. Here, we attempt this investigation using terahertz pulses centred at 3 THz with a bandwidth of 0.5 THz, generated with a tabletop source, as described in the Methods. The circular polarization is obtained with a quarter-wave plate designed for the 3 THz (100 μm) radiation. The ferroelectric phonon mode is driven along two of the three degenerate eigenvectors with the same phase shift, resulting in circular ionic motion. The circular nature of the polar phonon implies a polarization

[1]Department of Physics, Stockholm University, Stockholm, Sweden. [2]Department of Molecular Sciences and Nanosystems, Ca' Foscari University of Venice, Venice, Italy. [3]Elettra-Sincrotrone Trieste S.C.p.A., Basovizza, Italy. [4]European Synchrotron Radiation Facility, Grenoble, France. [5]Department of Physics and ISC-CNR, 'Sapienza' University of Rome, Rome, Italy. [6]Research Center for Magnetic and Spintronic Materials, National Institute for Materials Science, Tsukuba, Japan. [7]Linac Coherent Light Source, SLAC National Accelerator Laboratory, Menlo Park, CA, USA. [8]NORDITA, Stockholm, Sweden. [9]Department of Physics, University of Connecticut, Storrs, CT, USA. [10]Rara Foundation – Sustainable Materials and Technologies, Venice, Italy. [11]These authors contributed equally: M. Basini, M. Pancaldi. ✉e-mail: stefano.bonetti@fysik.su.se

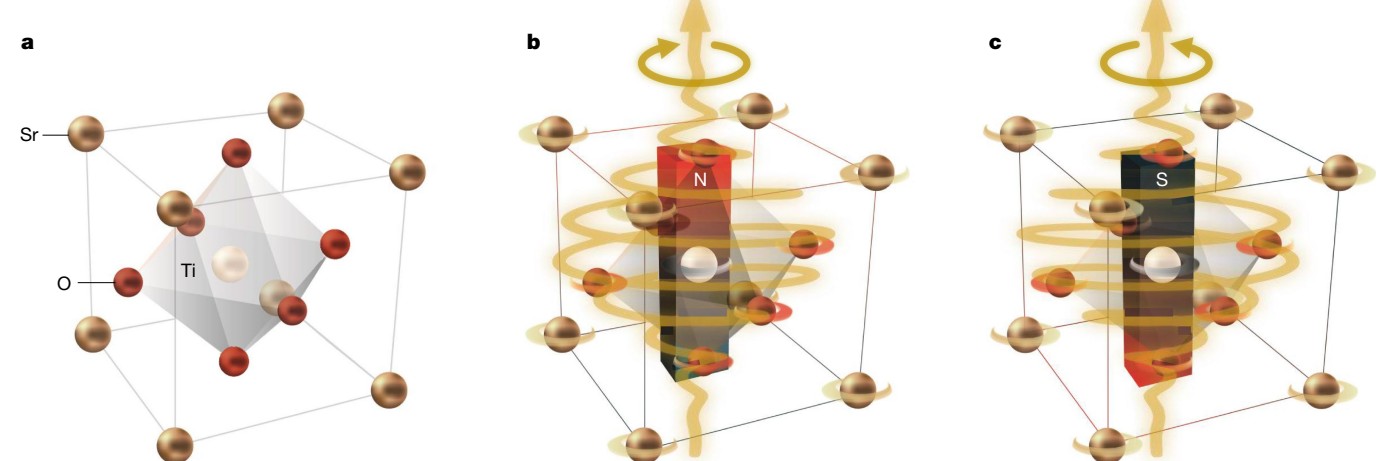

**Fig. 1 | Schematic of the experimental realization of dynamical multiferroicity. a**, SrTiO$_3$ unit cell in the absence of a terahertz electric field. When a circularly polarized terahertz field pulse drives a circular atomic motion, dynamical multiferroicity is expected to create a net magnetic moment in the unit cell. **b,c**, The north pole points up for a pulse that is left-handed (**b**), and down for a pulse that is right-handed (**c**).

**P** orthogonal to its time derivative $\partial_t$**P** during the whole dynamics, thus maximizing the cross product that is expected to give rise to the magnetization. To probe possible magnetic signals, we used the time-resolved magneto-optical Kerr effect, which measures the rotation of the polarization of a probe pulse reflected by a magnetized material[20]. The magneto-optical Kerr effect has been shown to be sensitive to an effective magnetic field driven by optical phonons in antiferromagnetic ErFeO$_3$ at low temperatures[21]. In that case, the amplification of an already existing coherent spin precession resonant with the magnetic field induced by the phonon motion was used as an indirect proof of the existence of such field. In the following, we show that magnetic order can be induced by dynamical multiferroicity also in non-magnetic systems in which there are no underlying magnetic excitations.

The temporal evolution of the Kerr rotation of a 400 nm, 40-fs-long probe pulse as a function of the time delay with respect to terahertz pump pulses of approximately 230 kV cm$^{-1}$ in amplitude is shown in Fig. 2a. We plot the difference signals measured using circularly polarized fields with opposite helicities. The individual measurements are shown in Extended Data Fig. 1. The signal exhibits one cycle of a slower oscillation at a frequency $\omega_-$ modulated by a faster oscillating component at a frequency $\omega_+$. In Fig. 2b we plot the calculated fast Fourier transform (FFT) of the time trace in Fig. 2a. This analysis clearly reveals a low-frequency component centred at approximately 0.6 THz and a faster one at around 6 THz. In the same figure, we also show the FFT of the product of the two components $E_x$ and $E_y$ composing the difference of the $z$-propagating circular pump field (centred at 3 THz, shown in Extended Data Fig. 2), retrieved with an independent electro-optical measurement (Methods). We normalize this product so that the peaks of the 6 THz components overlap, and we keep an identical approach when evaluating the theoretical response. Note that, already, by aligning the high-frequency peaks, there is a different amplitude in the low-frequency peaks, with the measured Kerr rotation being larger than the product of the two field components.

In Fig. 3a we plot the FFT of the measured difference Kerr rotation as a function of temperature, from 160 to 360 K, a temperature range large enough to move the soft phonon in and off resonance with the driving terahertz field. Note that the overall response is larger at a temperature of 280 K, and it decreases as the temperature moves away from it, both at lower and higher temperatures. Figure 3b shows the amplitude of the two FFT components of the measured Kerr rotation at $T$ = 300 K as a function of the applied terahertz electric field amplitude. For both frequency components, a clear quadratic dependence is observed, with a larger curvature for the lower frequency mode at $\omega_-$ than for the higher frequency one at $\omega_+$.

We next discuss the possible origin of the two peaks in the difference signal. Their frequency is consistent with the prediction of dynamical multiferroicity[7], with a peak $\omega_+$ at twice the soft phonon frequency and one at $\omega_- \approx 0$. However, from the dynamical multiferroicity theory[7], one would expect the peak at $\omega_+$ to have negligible amplitude, whereas the experimental data show that they are approximately equal.

Two peaks of comparable amplitudes are, on the other hand, expected when accounting for the electronic nonlinear response associated with $\chi^{(3)}$, namely the third-order susceptibility tensor[22]. We present a detailed combined theoretical and experimental study on this aspect in a separate work[23], which accounts for the symmetry properties of the crystal and for the specific spectral features of the pump pulse. We anticipate here that the theory predicts an effect proportional to the product of the $E_x$ and $E_y$ components of the incident terahertz field, as we show in Fig. 2b and in the Methods. Briefly, the product of the two field components at a frequency $f$ gives rise to a signal probed by the near-infrared field at $2f$ and to a rectified signal that is different from zero because of the finite bandwidth of the laser pulse. However, also note that although the shape and amplitude of the peak at $\omega_+$ can be reproduced exactly by the calculations based on the symmetry of the $\chi^{(3)}$ tensor, the experimental peak at $\omega_-$ is higher than what those same calculations predict and it cannot be accounted for with a different normalization scheme. The stronger quadratic dependence of the $\omega_-$ mode on the driving terahertz field, compared to the one for the $\omega_+$ mode and to the quadratic dependence of the $\omega_-$ mode for the $E_xE_y$ product, also supports the presence of an underlying extra contribution to the overall sample response.

It is important to stress that in the original dynamical multiferroicity theory[7], no anharmonic effects were taken into account. On the other hand, the phonon frequency and polarization vectors of STO are known to be strongly dependent on temperature due to lattice anharmonicities[24,25]. Therefore, for a quantitative description of the phenomenon, we calculate the effective phonon frequencies and polarization vectors renormalized by the quartic anharmonicity using self-consistent phonon (SCP) theory[9], as described in the Methods. We use these calculations to estimate the potential energy landscape for the two orthogonal displacement directions of the soft phonon. Then, we implement dynamical multiferroicity theory to compute the expected induced magnetization, which we plot in Fig. 4a,b in the time and, respectively, frequency domains. We also show the calculated

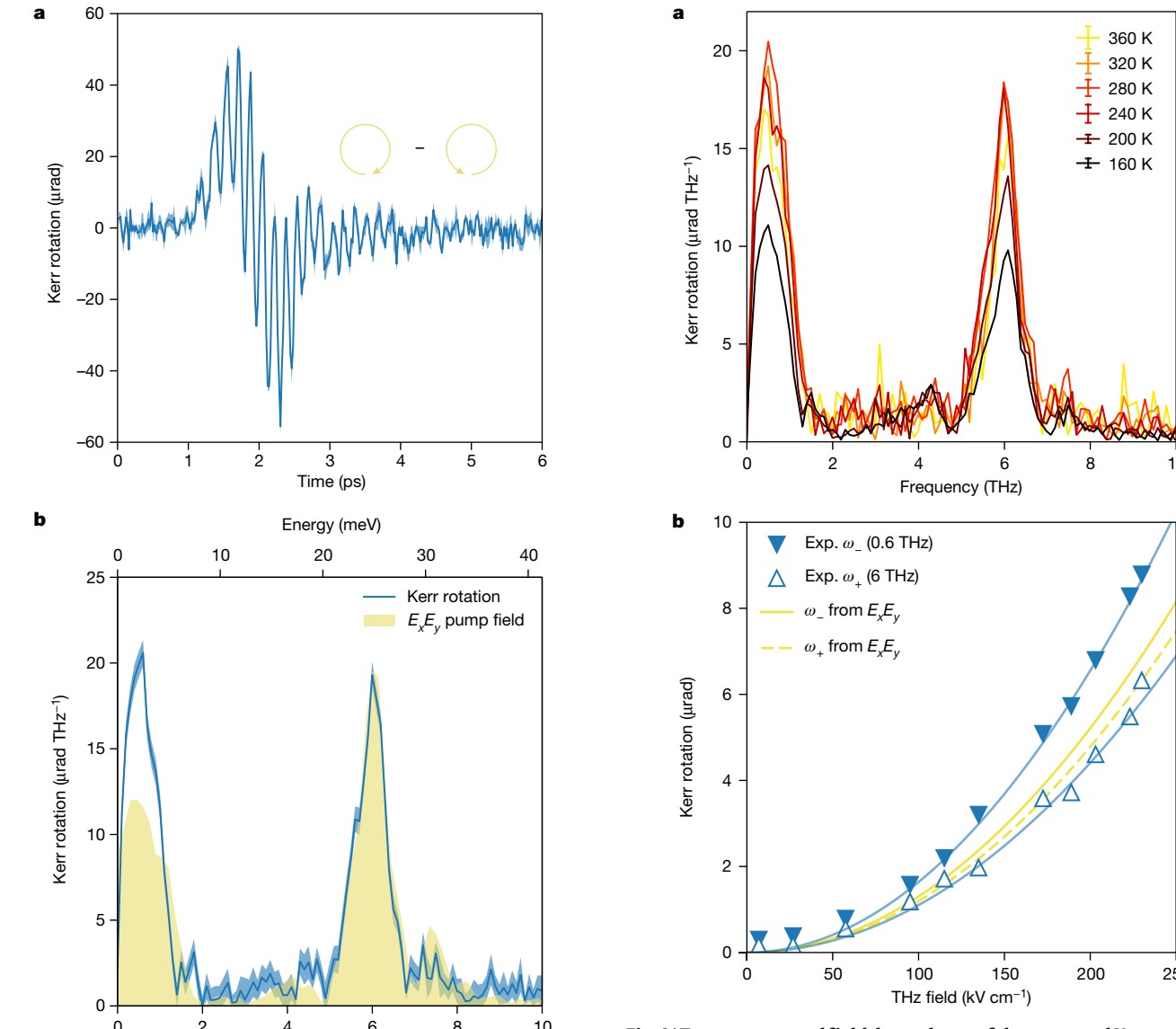

**Fig. 2 | Experimental detection of the time-resolved Kerr rotation.**
**a,b,** Measured Kerr rotation as a function of pump–probe delay for 400 nm probe pulses after excitation by the terahertz pump field. The probe polarization is parallel to the [100] crystal axis and the sample is tilted at an angle of 45° (Methods). **a,** Difference between the responses from circularly polarized fields with opposite helicities. **b,** FFT of the time trace in **a** together with the spectrum of the product of the two components $E_x$ and $E_y$ of the incident circular pump field (Methods). The standard error is indicated by the shaded area.

**Fig. 3 | Temperature and field dependence of the measured Kerr rotation.**
**a,** FFT of the difference of the magneto-optical Kerr effect response from opposite circularly polarized fields at a few selected temperatures. The error bars in the legend represent the standard error of the corresponding dataset. **b,** Dependence of the amplitude of the two main spectral peaks on the terahertz electric field. The standard error is smaller than the symbol size. Blue solid lines are quadratic fits to the data at $\omega_-$ and $\omega_+$. Yellow lines indicate the field dependence of the $\omega_-$ and $\omega_+$ peaks for the $E_x E_y$ product, after normalization to the experimental (Exp.) $\omega_+$ value at maximum field.

nonlinear optical response of the material due to the Kerr effect[23], which comprises an electronic (EKE) and a ionic (IKE) contribution, as described in the Methods. Finally, the expected polarization rotation resulting from the sum of the two effects is shown in the same figure. We used an identical normalization scheme as for the experimental data to match the amplitude of the 6 THz mode. We used the computed parameters for all quantities, except for the soft phonon eigenfrequency and its linewidth, for which experimental data are available[17,18] and are consistent with our own experimental observations. The computed magnetic moment is of the order of $10^{-2}\mu_N$ per unit cell, much smaller than the one computed in ref. 8 but consistent with a more realistic atomic displacement (approximately 1 pm for the Ti and Sr atoms and 3 pm for the oxygen ones), comparable to the one observed experimentally in ref. 19.

The resemblance of Fig. 4a,b to Fig. 2a,b is remarkable, given that no free adjustable parameters were used in our calculations. In particular, note that by combining the nonlinear optical response and the dynamical multiferroicity calculations, we obtain two frequency peaks with relative weights matching the experimental data. To investigate this further, we plot in Fig. 4c the expected magnetic moment induced by the dynamical multiferroicity as a function of temperature, again with no free adjustable parameters, and compare it to the measured extra weight of the low-frequency peak at different temperatures. The qualitative agreement between calculations and experiment in the temperature-dependent behaviour is outstanding. For example, there is a peaked response close to the temperature at which the soft phonon is resonant with the driving pump terahertz field.

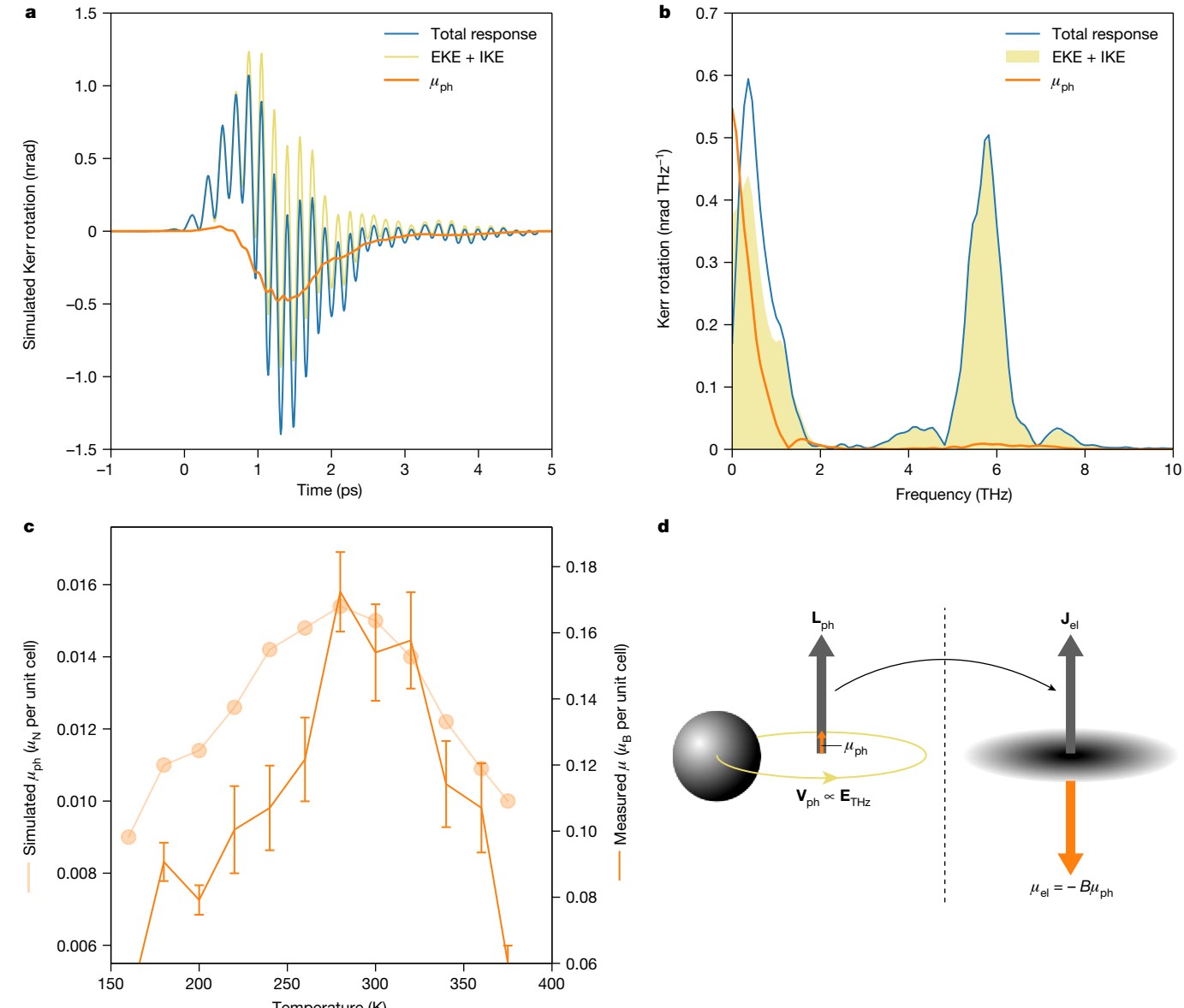

**Fig. 4 | Experimental and modelled dynamical multiferroicity. a**, Calculated total time-domain polarization rotation (blue), which includes the contribution of both EKE and IKE (yellow) and of the magnetic moment $\mu_{ph}$ due to dynamical multiferroicity (orange). **b**, FFT of the time-domain response, decomposing the spectral features of the different contributions. **c**, Calculated temperature dependence of the magnetic moment expected from the pure dynamical multiferroicity mechanism (semi-transparent symbols and line) and experimentally extracted spectral weight of the low-frequency peak (solid symbols and line). The error bars represent the standard error of the corresponding dataset. **d**, Pictorial representation of the phonon Barnett effect with enhancement factor $B$.

Remarkably, the estimated experimental magnetic moment—calculated using the measured Verdet constant for STO, the measured Kerr rotation and the estimated penetration depth of the terahertz radiation (Methods)—is approximately four orders of magnitude larger than the theoretical one, $10^{-1}\mu_B$ versus $10^{-2}\mu_N$ per unit cell. This suggests that, although dynamical multiferroicity can induce an effect qualitatively consistent with our observations, there mut be an additional and key mechanism to explain it quantitatively. We rule out enhancements of the magneto-optical signal due to microscopic mechanisms like those reported in ref. 26. Both that work and ours deal with effects having the symmetry of inverse Faraday effects; however, there are many such effects with the same phenomenological symmetry but with a different microscopic origin, as in our case.

Recently, a phonon magnetic moment four orders of magnitude larger than expected from pure phonon motion was observed in a Dirac semimetal[27]. In another work[28], the measured $g$ factor in PbTe related to a soft phonon at terahertz frequencies was found to be three orders of magnitude larger than predicted by theory. A recent theoretical work[29] found that in semimetals and small-gap (tens of millielectronvolts) insulators, a four orders of magnitude enhancement of the phonon magnetic moment can be expected[30–32]. None of these results apply directly to our sample, but they all reflect a similarly enhanced magnetic moment when phonons are involved. Finally, ab initio calculations have also shown that in STO the coupling strength between the electronic system and the soft phonon at the zone centre is very large, of the order of 1 eV (ref. 25). We stress, however, that the fundamental physical observable of relevance in this case is not the energy but the angular momentum.

All these observations point to an important gap in our understanding of the transfer of angular momentum from the phonon to electrons.

It has recently been suggested that in coupled electron–phonon dynamics the reduced mass of the system should be considered[33]. This would intuitively result in a magnetic moment enhanced by a factor of the order of the ratio of the masses of the proton and electron, owning to the larger gyromagnetic ratio, as we show pictorially in Fig. 4d. For our case, we observe an enhancement like the ratio $2m_p/m_e \approx 4 \times 10^3$, where the factor 2 accounts for the contribution of a neutron for each proton to the nuclear mass. We propose that the observed enhancement of the magnetic moment can be thought of as a microscopic version of the Barnett effect, which is the reciprocal of the Einstein–de Haas effect, in which a rigid rotation induces a magnetization in the rotating body. In our case, the body is not rigidly rotating, but there is coherent phonon motion that induces a transient net angular momentum in the lattice. We suggest keeping the same name, since a phonon Einstein–de Haas effect has been proposed theoretically[34–36] and recently observed experimentally[37,38]. On the other hand, an experimental realization of the phonon Barnett effect—the transfer of mechanical angular momentum from rotating phonons to the electronic system—has not been reported nor a scheme to detect it yet devised. We stress that with the current data, we can assume only that the transfer of angular momentum is between the orbital phonon angular momentum $\mathbf{L}_{ph}$ and the total electronic angular momentum $\mathbf{J}_{el}$. The present results cannot resolve between orbital and spin contributions; the latter could possibly arise if the strong circularly polarized terahertz field modifies the band structure to such an extent that spin-polarization is created and mediated by spin–orbit coupling, but large signals could be expected due to orbital contributions[39]. An alternative explanation was proposed during the reviewing process of this paper, which suggested that circularly polarized light can induce non-Maxwellian fields[40] that could explain the data. We leave these considerations for future works.

## Conclusion and outlook

We finally comment on the implications of our results, which are expected to be general for the several materials listed in ref. 8 with infrared-active phonons. Since many of these materials are commonly used as substrates, one could conceive of new ways to drive magnetic states at interfaces. A recent work[41], which appeared after the submission of this manuscript, shows this approach: circular phonons induced by circularly polarized light in a substrate are able to switch the magnetization of a ferromagnet on top of it. This strongly supports the observation that the induced magnetic moment must be of the order of the electronic one. At low temperatures, intense coherent terahertz excitation of the soft mode in STO can lead to highly nonlinear phonon responses that overcome the quantum fluctuations to create a ferroelectric order absent at equilibrium[42,43] that is coupled to the induced magnetic order. It is worth stressing that the induced magnetic moment can be seen as a quasi-static magnetization created on a timescale of a picosecond. This is about one order of magnitude faster than the fastest spin switching reported to date[44]. The applicability of the optical generation and control of magnetization by circular phonons can also be extended to two-dimensional materials, such as transition metal dichalcogenides. These materials have recently been predicted and observed to host chiral phonons that are intrinsically circularly polarized[45,46]. Finally, we anticipate that our results will stimulate further research on the microscopic understanding of angular momentum transfer at ultrafast and atomic scales and help in further exploring and understanding intriguing entangled orders in condensed matter[47–50].

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

## Methods

### Sample details

The sample considered was a 10 mm × 10 mm, 500-µm-thick $SrTiO_3$ crystal substrate from MTI Corporation, with the [001] crystallographic direction normal to the cut direction. Both sides were polished. The sample was mounted tilted at 45° with respect to the free space propagation direction to measure the geometry through reflection. However, the large dielectric constant ($|\tilde{\varepsilon}| \approx 100$) of STO at the resonant frequency of the soft phonon mode causes such a large refraction of the pump terahertz beam that the propagation within the crystal is always orthogonal with respect to the sample surface.

### Experimental methods

Broadband single-cycle terahertz radiation was generated by optical rectification in a DSTMS (4-$N$,$N$-dimethylamino-4′-$N$′-methyl-stilbazolium 2,4,6-trimethylbenzenesulfonate) crystal[15] of a 40-fs-long, 800 µJ near-infrared laser pulse centred at a wavelength of 1,300 nm. This pulse was generated by optical parametric amplification from a 40-fs-long, 6.3 mJ pulse centred at 800 nm wavelength, which was produced by a 1 kHz regenerative amplifier. The terahertz pulses were focused onto the sample by three parabolic mirrors to a rounded beam of approximately 0.5 mm in diameter. Knowing the exact size of the beam is not crucial when estimating the fluence as we characterized the electric field of the radiation. A pair of wire-grid polarizers was used to tune the field amplitude without affecting the pulse shape. Narrowband terahertz radiation was obtained by filtering the broadband field with a 3 THz band-pass filter resulting in a peak frequency of 3 THz and a full-width at half-maximum of 0.5 THz. The probe beam was a 40-fs-long pulse at 800 nm wavelength, produced by the same 1 kHz regenerative amplifier used to generate the pump radiation. A β-barium borate crystal was used to convert the probe wavelength to 400 nm. The probe polarization was set by a nanoparticle linear film polarizer. The probe size at the sample was approximately 100 µm in diameter, substantially smaller than the terahertz pump. To record the change in the polarization state of the probe beam, a Wollaston prism was used to implement a balanced detection scheme with two photodiodes. A half-wave plate was used before the Wollaston prism to detect the Kerr rotation. The signals from the photodiodes were fed to a lock-in amplifier, whose reference frequency (500 Hz) came from a mechanical chopper mounted along the pump path.

### Characterizing the terahertz electric field

The electric field component of the terahertz pulse at the sample location was characterized by electro-optical sampling[51] in a 50-µm-thick (110)-cut GaP crystal. In particular, the field strength was calculated using a standard procedure, namely by measuring the time-resolved birefringence at a wavelength of 800 nm, which was caused by the terahertz electric field in the GaP crystal. To do that, we used a balanced detection scheme with two photodiodes measuring the $I_1$ and $I_2$ signals produced by a quarter-wave plate and a Wollaston prism placed after the sample. Once the time trace was retrieved, we moved the delay stage to the time delay where the maximum of the terahertz electric field was found. Following ref. 52, we computed $E_{THz} = \lambda_0 \sin^{-1}((I_1 - I_2)/(I_1 + I_2))/(2\pi n_0^3 r_{41} t_{GaP} L)$, where $n_0 = 3.193$ is the refractive index of GaP at 800 nm, $L = 50$ µm is the thickness of the crystal, $\lambda_0 = 800$ nm is the probe wavelength, $r_{41} = 0.88$ pm $V^{-1}$ is the GaP electro-optic coefficient[53] and $t_{GaP} = 0.4769$ is the Fresnel coefficient for reflective loss at the GaP crystal. For the DSTMS-generated broadband pulse, the maximum measured terahertz peak electric field was approximately 1.15 MV $cm^{-1}$, and the peak frequency of the terahertz pump pulse was at 2.7 THz with measurable components extending up to approximately 5 THz. After filtering the field with the 3 THz band-pass filter, a typical measured terahertz peak electric field was around 200–300 kV $cm^{-1}$. The sampled terahertz pump traces are reported in Extended Data Fig. 2, which shows the narrowband data representing the field used to measure all the data in the main text.

### Circular polarization of the terahertz beam

After filtering the broadband terahertz radiation with a terahertz band-pass filter, the linear polarization can be converted into circular polarization with a terahertz quarter-wave plate oriented at ±45° to obtain the opposite helicities. For this purpose, we chose a Tydex quarter-wave plate made of x-cut terahertz-grade crystal quartz, whose thickness was adjusted to provide a π/2 phase shift at 3 THz. However, for wave plates, the phase shift is very sensitive to the radiation frequency. Moreover, we were not working with a monochromatic beam. Nonetheless, in our case the use of a quarter-wave plate was justified because the bandwidth of the filtered pulse was narrow enough to allow the wave plate to operate according to its design. To characterize the polarization state of the circular pump beam, we measured both the $E_x$ and $E_y$ electric field components with electro-optical sampling by rotating the GaP crystal by 90° around the light propagation axis. This allowed us to get the sensitivity to two orthogonal components of the terahertz pump, as shown in ref. 51. The time-domain traces obtained are shown in Extended Data Fig. 2 for the two helicities, labelled as LCP and RCP. From these traces, the polarization state can be unambiguously identified by calculating the Stokes parameters in the frequency domain, as shown in Extended Data Fig. 3. The $S_3$ parameter is associated with circular polarization, and a change of sign represents opposite helicities. The following inequality holds for a broadband pulse[54]: $(S_1^*/S_0^*)^2 + (S_2^*/S_0^*)^2 + (S_3^*/S_0^*)^2 \leq 1$, where $S_0^* = \sum_i S_{0,i}$, $S_1^* = \sum_i S_{1,i}$, $S_2^* = \sum_i S_{2,i}$, $S_3^* = \sum_i S_{3,i}$ and $i$ represents the $i$th frequency. The $(S_3^*/S_0^*)^2$ quantity gives an indication of the average amount of circular polarization in the terahertz pump pulse. Considering only the 0.5 THz full-width at half-maximum region of the peak, we found that the beam was 85–90% circularly polarized, as summarized in Extended Data Fig. 3.

### Evaluating the complex refractive index

The complex refractive index $\tilde{n} = n + ik$ of $SrTiO_3$ was derived from a combination of previous ellipsometry measurements on STO thin films[55] and hyper-Raman scattering in bulk STO[18]. In particular,

$$n = \sqrt{\frac{|\tilde{\varepsilon}| + \varepsilon_1}{2}}, \qquad k = \sqrt{\frac{|\tilde{\varepsilon}| - \varepsilon_1}{2}},$$

where $\tilde{\varepsilon} = \varepsilon_1 + i\varepsilon_2$ is the complex permittivity. To estimate the permittivity in the experimental temperature range 160 K < $T$ < 375 K for our specific sample, we first used the experimental data of ref. 55 at $T = 300$ K (Extended Data Fig. 4), which contains a broadband response that can be fitted with a Lorentz oscillator. Then, to adjust it to our case, we rigidly shifted the curve, moving the peak from 3 to 2.7 THz, in accordance with our own data and ref. 17 on bulk samples. All centre frequency and linewidth values at the different temperatures are listed in Extended Data Table 1. All values for $n$ and $k$ are reported in Extended Data Fig. 4.

### Modelling the terahertz reflectance, transmittance and absorptance

The electric field reflection, absorption and transmission properties were calculated for an air/STO/air stack using the analytical formulas for optical trilayers at normal incidence[56]:

$$r = \frac{C - C\exp(2i\delta)}{1 - C^2\exp(2i\delta)}, \qquad t = \frac{(4\tilde{n}/(1+\tilde{n})^2)\exp(i\delta)}{1 - C^2\exp(2i\delta)},$$

$$C = \frac{1-\tilde{n}}{1+\tilde{n}}, \qquad \delta = \frac{2\pi d}{\lambda}\tilde{n},$$

where $\tilde{n}$ is the refractive index of STO, $d$ is the thickness of the STO sample, $\lambda$ is the wavelength and the refractive index of air is considered

to be 1. The reflectance, transmittance and absorptance are given, respectively, by $R = |r|^2$, $T = |t|^2$ and $A = 1 - R - T$. Considering $d = 500\ \mu m$, $\lambda = 100\ \mu m$ (3 THz) and $\tilde{n} = 3.8 + i6.4$ from ref. 17 at a temperature of 300 K, we get $R \approx 0.76$, $T \approx 0$ and $A \approx 0.24$. As $T \approx 0$, it is also interesting to estimate the decay length $l_{\text{decay}}$ of the electric field inside STO, which indicates how much of the pump radiation penetrates into the sample:

$$l_{\text{decay}}(\tilde{n}) = \frac{\lambda}{2\pi\,\Im(\tilde{n})} \approx 2.49\ \mu m.$$

The estimated penetration depth in the experimental temperature range 160 K < $T$ < 375 K is listed in Extended Data Fig. 4.

**Estimating the polarization rotation and magnetic field**
Measuring the probe polarization rotation allowed us to calculate the magnetic field induced in STO. According to theory, the Faraday rotation $\vartheta_F$ and magnetic field are connected through the equation[20]

$$\vartheta_F = VB\int_0^d \exp\left(-2\frac{z}{l_{\text{decay}}}\right) dz = VB\frac{l_{\text{decay}}}{2},$$

as $l_{\text{decay}} \ll d$, where $d$ is the STO thickness. The parameter $B$ represents the amplitude of the magnetic field at the surface, $l_{\text{decay}}$ is the decay length of the pump field and $V$ is the Verdet constant. The factor of 2 in the exponential function appears because the induced magnetic field is proportional to the square of the pump electric field. To extract the magnetic moment generating $\vartheta_F$, we exploited the relation $B = \mu_0 M$, where $M$ is the magnetization induced by the pump. Considering the STO lattice parameter $a = 3.9$ Å, the magnetic moment per unit cell $\mu$ is given by

$$\mu = \frac{2\vartheta_F a^3}{\mu_0 V l_{\text{decay}}}.$$

Even if the measurements reported in the main text are performed in reflection (Kerr rotation $\vartheta_K$), we evaluated the magnetic field considering a transmission measurement (Faraday geometry). Those were the only reliable values for the Verdet constant that we could find, which we were able to validate ourselves, as shown in Extended Data Fig. 5. To confirm that the reflection and transmission geometries give comparable responses, we measured the Faraday rotation during the same set of experiments described in the main text. We found that the absolute measured signal is within a factor of 2 compared to the Kerr rotation. Moreover, the pump penetration depth was still the limiting factor for the decay length $l_{\text{decay}}$ to be considered in the above equation, as even in reflection, the STO thickness probed by the probe pulse is more than the pump penetration depth. The thickness contributing to the probe signal in reflection for an ultrafast pulse can be estimated through the distance travelled in the material during the pulse duration. For a 400 nm probe pulse ($n_{\text{STO}} = 2.6$ from ref. 57) of 50 fs duration, the distance travelled in the STO during that interval corresponds to approximately 5.8 μm, which is longer than all the pump field penetration depths listed in Extended Data Fig. 4. According to ref. 58, we have $V \approx 250$ rad m$^{-1}$ T$^{-1}$, so that for $\vartheta_K = 10$ μrad and $l_{\text{decay}} = 2.49$ μm, the magnetic field at the surface $B \approx 0.032$ T. The average energy $\epsilon$ stored per unit surface in such a magnetic field is given by:

$$\epsilon = \frac{1}{2}\frac{1}{2\mu_0}B^2\int_0^d \exp\left(-4\frac{z}{l_{\text{decay}}}\right) dz = \frac{1}{16\mu_0}B^2 l_{\text{decay}} \approx 0.013\ \mu J\ cm^{-2},$$

where the first factor of 1/2 in the definition of $\epsilon$ is due to the time average of the square of a sine wave, as we approximate the slow oscillation in Fig. 2a with a sinusoidal function. The integral takes into account that the induced magnetic field does not fill the whole sample volume but has a finite penetration depth. The energy $\epsilon$ is delivered by the pump

pulse, and its fluence can be calculated by integrating the square of the trace shown in Extended Data Fig. 2 to give approximately 60 μJ cm$^{-2}$, which is much higher than the energy per unit surface in the generated magnetic field.

The pump fluence can be used to compute the absorbed energy density and give an estimate of the related temperature variation. As stated above, at 300 K, the decay length of the terahertz pump electric field is 2.49 μm and the absorptance is 0.24, which leads to an estimate of the average energy density absorbed by the sample of 115.7 mJ cm$^{-3}$ = 0.043 meV per unit cell. The temperature increase for such an energy density can be obtained from the heat capacity and density of STO. Considering a density of 5.18 g cm$^{-3}$ (MTI Corporation), a heat capacity of 100 J K$^{-1}$ mol$^{-1}$ (ref. 59) and a molar mass of 183.5 g mol$^{-1}$, the temperature increase is expected to be approximately 0.04 K, which could be neglected during the temperature-dependent measurements presented in the main text.

We also checked that the measured magneto-optical effect was not affected by the probe wavelength being too close to the bandgap of the material. In Extended Data Fig. 6, we present the total measured Faraday effect at both 400 and 800 nm probe wavelengths. Apart from an overall scaling factor consistent with the different values of the Verdet constant, the scaled response is identical for the two wavelengths, excluding wavelength-dependent artefacts.

**Modelling the total Kerr effect**
In ref. 23 it was shown that the EKE response is given by

$$\Delta\Gamma^e \propto \frac{1}{4}[E_x^2 - E_y^2]\Delta\chi \sin(4\vartheta) + 2E_x E_y\left[\chi_{iijj}^{(3)} + \frac{1}{2}\Delta\chi \sin^2(2\vartheta)\right],$$

where $E_x$ and $E_y$ are the components of the pump pulse along generic $x$ and $y$ orthogonal directions, $\vartheta$ is the angle that $x$ and $y$ form with respect to the main crystallographic axes ($i$ and $j$) and $\Delta\chi \equiv \chi_{iiii}^{(3)} - 3\chi_{iijj}^{(3)}$, as $\chi_{iijj}^{(3)} = 0.47\chi_{iiii}^{(3)}$ are the only two independent tensor components of the $\chi^{(3)}$ tensor in cubic STO from ref. 58. If $\vartheta = 45°$, then $\Delta\Gamma^e \propto 2E_x E_y\left[\chi_{iijj}^{(3)} + \frac{1}{2}\Delta\chi\right]$ and the signal is proportional to the product of the terahertz pump field components along perpendicular directions. For circularly polarized light of opposite helicities (left and right), the signal difference $\Delta\Gamma^e$ (LCP) − $\Delta\Gamma^e$ (RCP) is still proportional to $E_x E_y$, as only one of the two pump components changes sign.

Besides the EKE, it has been shown that an additional contribution, IKE, associated with the nonlinear excitation of the infrared-active soft phonon mode, is present[23]. The IKE response $\Delta\Gamma^{ph}$ can be effectively modelled by replacing the $E_x$ and $E_y$ components in $\Delta\Gamma^e$ with a convolution between the pump and the single- or two-phonon propagators to account for the intermediate second-order excitation of the soft mode. Moreover, the $\chi^{(3)}$ tensor should be replaced with an effective nonlinear coupling between the pump and probe pulses and the infrared-active phonon.

An ab initio estimation of the effective nonlinear coupling is needed to estimate the relative weights of the IKE and the EKE in a rigorous way. This would require a state-of-the-art extension of the available density functional theory (DFT) codes, which has been investigated only recently[60] and goes far beyond the scope of this work. For this reason, in the main text we decided to model the full Kerr response with only the electronic contribution by assuming that the ionic contribution has a similar spectral content, so as not to introduce any free adjustable parameter into our simulations. For completeness, the full Kerr effect, including both the EKE and the IKE, is shown in Fig. 4a,b. The relative weight between the electronic and ionic contributions has been fixed to better reproduce the experimental time traces. This was done using the Kerr response measured with linearly polarized terahertz pulses as a reference.

Extended Data Fig. 7 compares the experimental and calculated responses of the material to linearly and circularly polarized terahertz

pump fields. These measurements allowed us to isolate the dependence of the response on the polarization that is beyond the one captured by the third-order susceptibility, in particular for the EKE description. In Extended Data Fig. 7a,b, we use the linearly polarized pump data to match the experimental and calculated amplitudes. With the same scaling factor applied to all the data, in Extended Data Fig. 7c, we plot the difference between the experimental and simulated Kerr rotation for the circularly polarized pump case. The part of the signal before the zero-crossing point at approximately 2.4 ps can be explained in terms of the IKE effect (which we can also model as discussed above but was left out for simplicity of reasoning), whereas the negative dip after the zero-crossing point is the signature of dynamical multiferroicity.

## Ab initio calculations

First-principles phonon calculations of cubic SrTiO$_3$ were performed within DFT using the Vienna Ab initio Simulation Package[61], which implements the projector augmented-wave method[62]. The adopted projector augmented-wave potentials treat Sr $4s^2 4p^6 5s^2$, Ti $3s^2 3p^6 4s^2$, and O $2s^2 2p^2$ as the valence states. An energy cutoff of 550 eV was used, and the Brillouin-zone integration was performed with a $12 \times 12 \times 12$ gamma-centred $k$-point mesh. The Heyd–Scuseria–Ernzerhof hybrid functional (HSE06; ref. 63) was adopted to give an accurate description of the phonon potential energy surface. The lattice constant was optimized within HSE06; the optimized value of 3.900 Å agrees well with the experimental value, 3.905 Å.

The effective phonon frequencies and eigenvectors at room temperature were calculated based on SCP theory, as implemented in the software ALAMODE (ref. 64). The harmonic and fourth-order interatomic force constants (IFCs), which are necessary as inputs to the SCP calculation, were calculated with the real-space supercell approach using a $2 \times 2 \times 2$ supercell. The harmonic IFCs were estimated by systematically displacing each atom in the supercell from its equilibrium site by 0.01 Å, calculating forces by DFT and fitting the harmonic potential to the displacement–force datasets. The fourth-order IFCs were estimated using the compressive sensing method, for which 40 training structures were generated by combining DFT and molecular dynamics with random displacements following ref. 65.

After obtaining the harmonic and anharmonic IFCs, the effective phonon frequency $\omega$ with branch index $\nu$ at wavevector $\mathbf{q}$ and the corresponding eigenvector were obtained by solving the SCP equation:

$$\omega_{\mathbf{q}\nu}^2(T) = (C_{\mathbf{q}}^\dagger \Lambda_{\mathbf{q}}^{HA} C_{\mathbf{q}})_{\nu\nu} + \frac{1}{2} \sum_{\mathbf{q}',\nu'} \Phi^{SCP}(-\mathbf{q}\nu; \mathbf{q}\nu; -\mathbf{q}'\nu'; \mathbf{q}'\nu')$$
$$\times \frac{\hbar(1 + 2n(\omega_{\mathbf{q}'\nu'}))}{2\omega_{\mathbf{q}'\nu'}},$$

where $\Lambda_{\mathbf{q}}^{(HA)} = \text{diag}(\widetilde{\omega}_{\mathbf{q},1}^2, ..., \widetilde{\omega}_{\mathbf{q},\nu}^2)$ with harmonic frequencies $\widetilde{\omega}$. $C$ is a unitary transformation matrix that modifies the polarization vector at finite temperature, $\Phi^{SCP}$ is the fourth-order anharmonic force constant and $n_{q,\nu}(\omega_{q,\nu})$ is the Bose–Einstein distribution. The equation was solved numerically by iteratively updating the effective frequency $\omega_{\mathbf{q}\nu}(T)$ and the unitary matrix $C_{\mathbf{q}}$ for the phonon modes at the gamma-centred $2 \times 2 \times 2$ $\mathbf{q}$ points.

The summation over the $\mathbf{q}'$ points was conducted with the denser $10 \times 10 \times 10$ $\mathbf{q}'$ points, which was sufficient to achieve convergence. The quartic coupling coefficient $\Phi^{SCP}(-\mathbf{q}\nu; \mathbf{q}\nu; -\mathbf{q}'\nu'; \mathbf{q}'\nu')$ was obtained from the fourth-order IFCs with the Fourier interpolation. The splitting of longitudinal optical and transverse optical modes was considered in the SCP calculation. The obtained SCP frequencies agree well with the inelastic neutron scattering data, as shown in Extended Data Fig. 8.

The anharmonic coupling coefficients of the triply-degenerate $\Gamma_{15}$ modes at room temperature were obtained by transforming the anharmonic IFCs into the normal coordinate basis. In this study, the anharmonic coupling terms up to the fourth-order were included in

$V(Q_1, Q_2)$. The normal coordinate $Q_\nu$ at finite temperature is given as $Q_\nu = \sum_\kappa \sqrt{m_\kappa} \mathbf{e}_\nu(\kappa) \cdot \mathbf{u}(\kappa)$, where $m_\kappa$ is the mass of atom $\kappa$ and $u^\alpha(\kappa)$ is its displacement in the $\alpha$ direction. The polarization vector at room temperature $\mathbf{e}_\nu(\kappa)$ was calculated as $\mathbf{e}_\nu(\kappa) = \sum_{\nu'} \widetilde{\mathbf{e}}_{\nu'}(\kappa)[C_{\mathbf{q}=0}]_{\nu'\nu}$, where $\widetilde{\mathbf{e}}_\nu(\kappa)$ is the harmonic polarization vector and $C_{\mathbf{q}}$ is the unitary matrix obtained as a solution to the SCP equation. As polarization mixing is significant in STO, the temperature dependence of the polarization vectors is noteworthy, as shown in Extended Data Table 2 for 300 K. As each atomic site of cubic STO is an inversion centre, all cubic coefficients became exactly zero. The effective charges of the $\Gamma_{15}$ modes were calculated as

$$Z_{\nu,\alpha}^* = \sum_{\kappa\beta} Z_{\kappa,\alpha\beta}^* \frac{e_\nu^\beta(\kappa)}{\sqrt{m_\kappa}},$$

with $Z_{\kappa,\alpha\beta}^*$ being the Born effective charge of atom $\kappa$. For 300 K, we have that $Z_{Si,\alpha\beta}^* = 2.553\delta_{\alpha\beta}$, $Z_{Ti,\alpha\beta}^* = 6.704\delta_{\alpha\beta}$, $Z_{O,\perp}^* = -1.941$ and $Z_{O,\parallel}^* = -5.375$, in units of the electron charge. The effective charge of the oxygen atom is different when considering the direction perpendicular ($\perp$) or parallel ($\parallel$) to the nearest titanium atom, and $\delta_{\alpha\beta}$ is the Kronecker delta.

## Phenomenological model for anharmonically coupled oscillators

To model the driven circular excitation of the ferroelectric mode along the two orthogonal directions, we derived the effective phonon potential for two of the threefold degenerate modes at $q = 0$, where the anharmonic coupling is included up to fourth order:

$$V(Q_1, Q_2) = \frac{1}{2}\omega^2 Q_1^2 + \frac{1}{2}\omega^2 Q_2^2 + \frac{1}{4}kQ_1^4 + \frac{1}{4}kQ_2^4 + \chi Q_1^2 Q_2^2 + \psi Q_1^3 Q_2 + \psi Q_1 Q_2^3,$$

where $Q$ is the normal coordinate in real space, the indices 1 and 2 refer to the two degenerate branches of the soft phonon along [100] and [010], $k$ is the anharmonic contribution to the potential, and $\chi$ and $\psi$ are the phonon–phonon coupling terms. As $Q_1$ and $Q_2$ are orthogonal to each other and the phonon potential spanned by them has a $C_4$ symmetry, $\psi = 0$. The resulting potential, with calculated parameters stated in Extended Data Table 1, represents two coupled anharmonic oscillators. The solution of this model is obtained by numerical integration of its equation of motion:

$$\ddot{Q}_i + \frac{\partial V}{\partial Q_i} + \Gamma \dot{Q}_i = Z^* \widetilde{E}_i^{THz}, \qquad i = 1, 2,$$

where $\Gamma$ accounts for the lifetime of phonons and $Z^* \widetilde{E}_i^{THz}$ is the oscillator coupling to the driving field through the mode effective charge $Z^*$. The effective field in the sample is expressed through the term $\widetilde{E}_i^{THz} = \alpha E_i^{THz}$ where $\alpha$ quantifies the amount of field actually experienced (not screened) by the sample. The value of $E_i^{THz}$ was fixed from our experiment, whereas the values of $\Gamma$ and $\omega$ at room temperature were taken from hyper-Raman measurements on bulk STO (ref. 17). Finally, the induced magnetic moment can be calculated via:

$$\mu = \gamma Q \times \dot{Q} = \sum_i \gamma_i Q_i \times \dot{Q}_i = \sum_i \gamma_i L_i,$$

where $i$ now represents the $i$th atom in the unit cell (Sr, Ti, O, O, O), $\gamma_i = eZ_i^*/2m_i$ is the gyromagnetic ratio and $L_i = Q_i \times \dot{Q}_i$ is the angular momentum. The calculated magnetic moment per unit cell $\mu$ is shown in Fig. 4a in the time domain and in Fig. 4b in the frequency domain using an approach identical to that used to process the experimental data. All parameters used to solve the equation of motion were fixed, except for $\alpha$, which was set to 0.7. The mode effective charge $Z^*$ and potential parameters $k$, $\chi$ and $\psi$ were calculated from first principles (Extended Data Table 1), whereas the excitation field $\widetilde{E}_i^{THz}$ and the phonon frequency $\omega$ and lifetime $\Gamma$ are those obtained in experiments[17].

# Data availability

Source data are provided with this paper.

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

**Acknowledgements** M.B. and S.B. acknowledge support from the Knut and Alice Wallenberg Foundation (Grant No. 2017.0158). M.B., V.U., A.V.B. and S.B. acknowledge support from the Knut and Alice Wallenberg Foundation (Grant No. 2019.0068). A.V.B. acknowledges a synergy grant from the European Research Council (Grant No. HERO-810451) and support from the University of Connecticut Quantum CT. M.U. acknowledges support from the European Union (Project MORE-TEM ERC-SYN under Grant Agreement No. 951215). T.T. acknowledges support from the Japan Society for the Promotion of Science (KAKENHI Grant No. 21K03424). We gratefully acknowledge discussions with M. Geilhufe, M. Fechner, D. Afanasiev, D. Bossini and L. Benfatto.

**Author contributions** S.B. conceived the project and designed the experiment and the model with contributions from M.B., M.P. and A.V.B. M.B., M.P. and S.B. analysed the data presented in this work. M.B., M.P., V.U. and S.B. performed the experiments and wrote the code to calculate the coupled driven dynamics. M.U. performed the simulations that calculated the nonlinear optical response. B.W. and T.T. designed and performed the ab initio calculations. M.C.H. performed additional independent measurements to confirm some of the experimental findings. All authors discussed the results. M.B. and S.B. wrote the manuscript with input from all authors.

**Funding** Open access funding provided by Stockholm University.

**Competing interests** The authors declare no competing interests.

**Additional information**
**Correspondence and requests for materials** should be addressed to S. Bonetti.

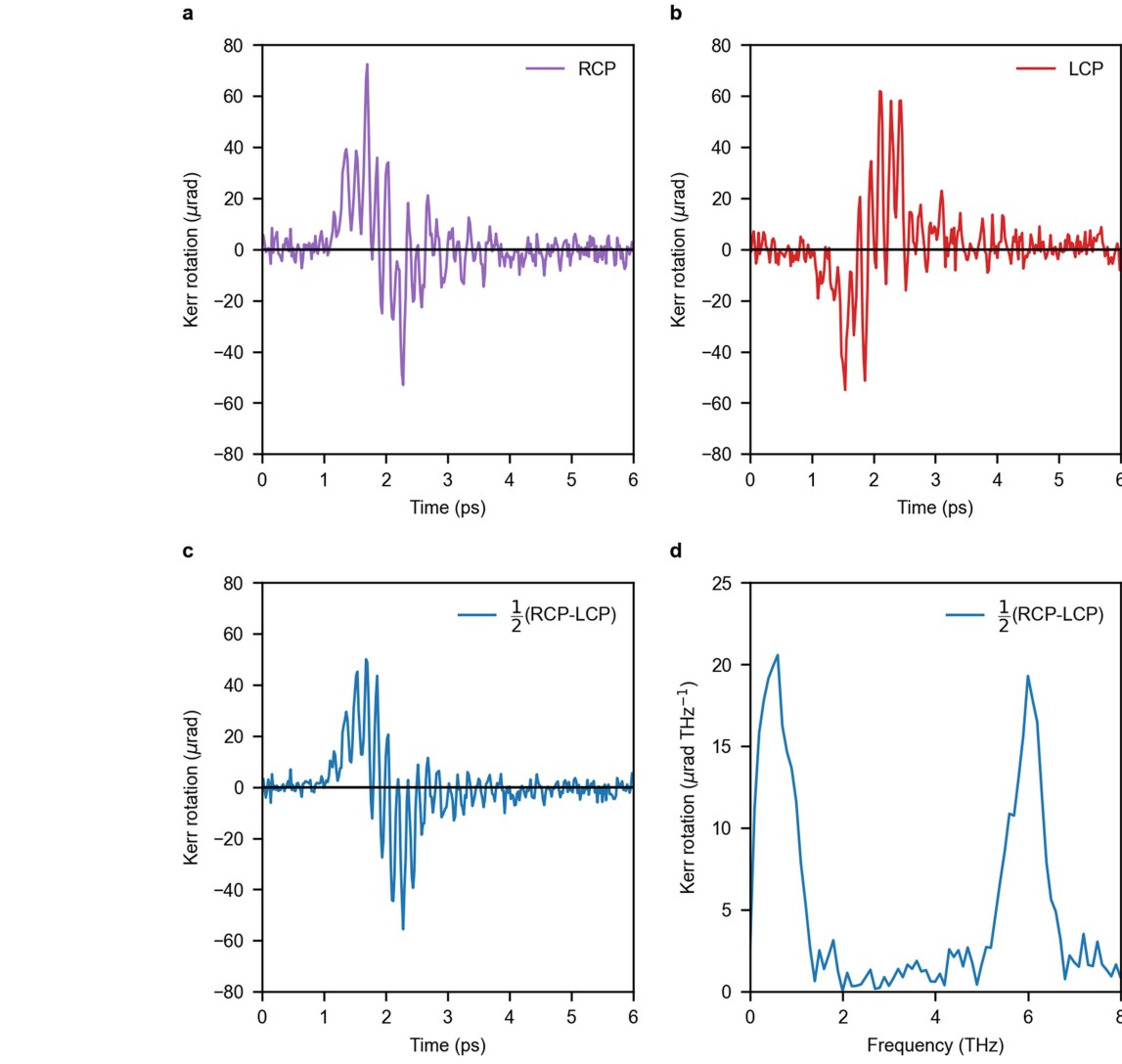

**Extended Data Fig. 1 | Individual helicity measurements and difference.** **a,b**, Measured Kerr rotation as a function of pump–probe delay for 400 nm probe pulses after sample excitation via right (RCP) (**a**) and left (LCP) (**b**) circularly polarized terahertz pump fields. **c**, Difference of the responses from circularly polarized fields with opposite helicities (RCP-LCP)/2. **d**, Its Fourier transform.

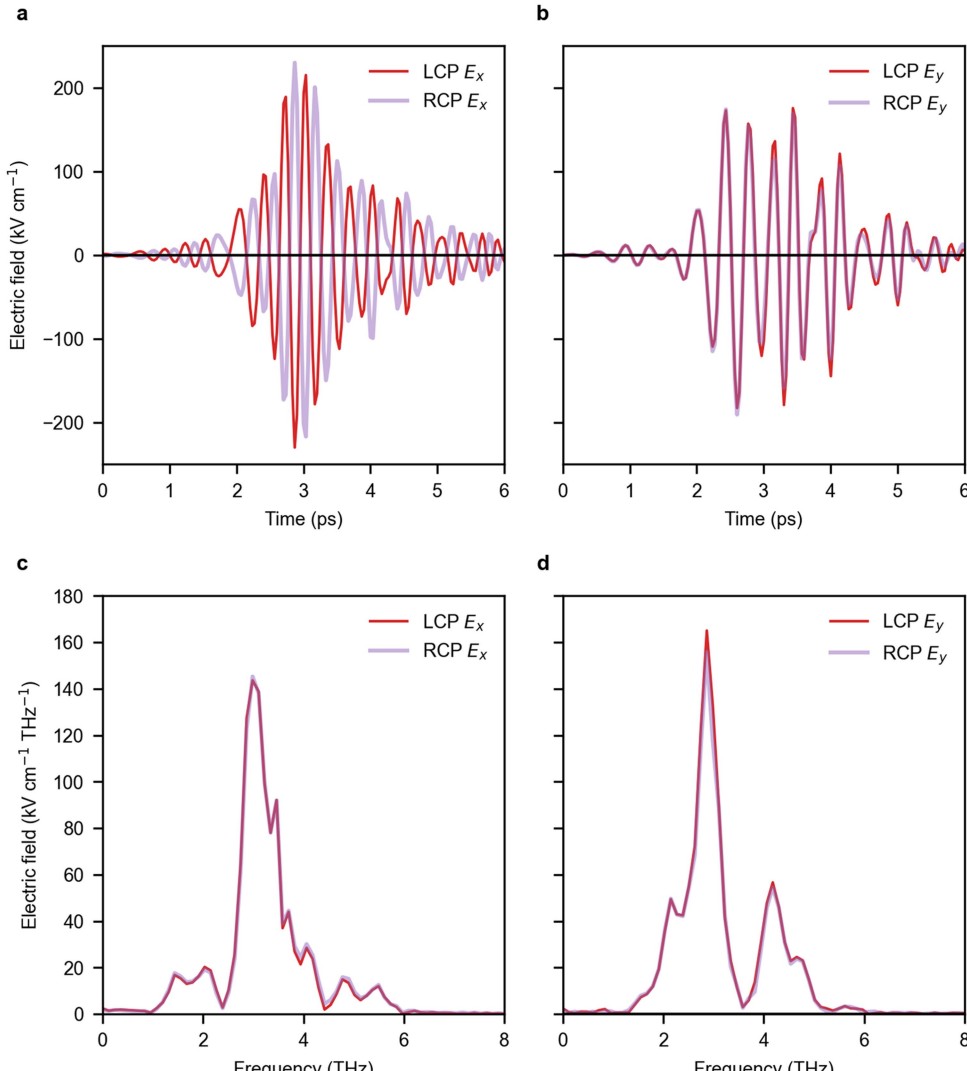

**Extended Data Fig. 2 | Electro-optical sampling data of the circularly polarized terahertz fields. a,b**, Recorded temporal traces. **c,d**, Their Fourier transforms. The measurements were performed using a 50-μm-thick GaP crystal cut along the 110 crystallographic direction, and the broadband pulse was filtered by means of a 3 THz filter with approximately 10% bandwidth, as described in the Methods. The two components $E_x$ and $E_y$ of the electric field are shown with red (left circular polarization, LCP) and violet (right circular polarization, RCP) solid lines. In **a,b**, the reflected terahertz field from the back side of the GaP crystal starts to interfere with the direct beam at approximately $t$ = 3.8 ps. This complicates the electro-optic sampling data presented here, but it has no importance in all measurements on the STO crystal presented in the main text. In STO, the large terahertz absorption suppresses the back-side reflection below the experimental noise level.

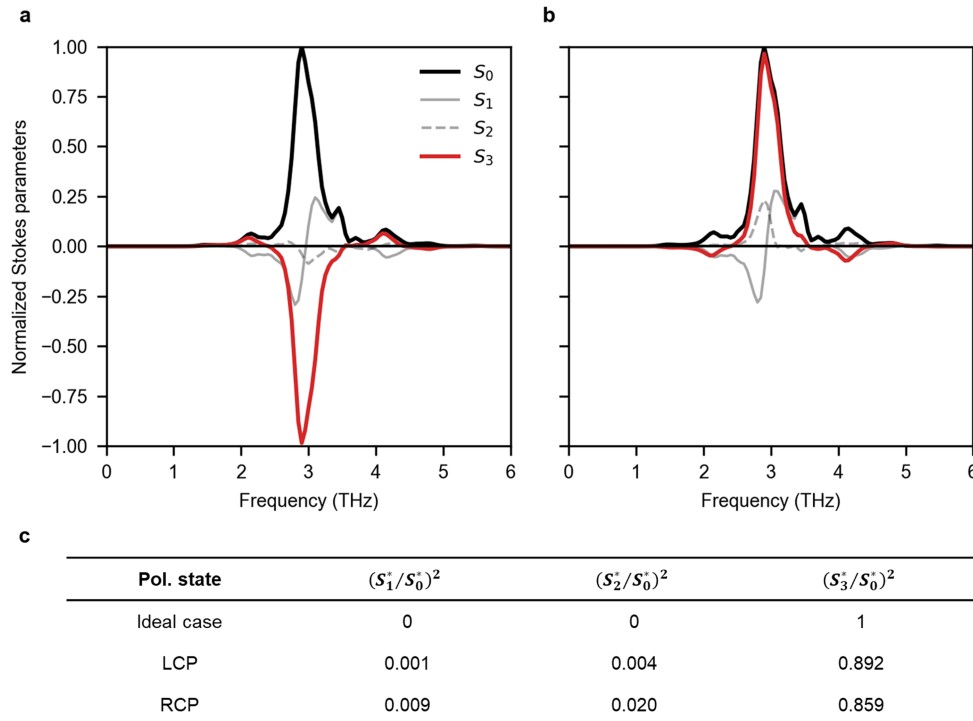

**c**

| Pol. state | $(S_1^*/S_0^*)^2$ | $(S_2^*/S_0^*)^2$ | $(S_3^*/S_0^*)^2$ |
|---|---|---|---|
| Ideal case | 0 | 0 | 1 |
| LCP | 0.001 | 0.004 | 0.892 |
| RCP | 0.009 | 0.020 | 0.859 |

**Extended Data Fig. 3 | Stokes parameters in the frequency domain. a**, Stokes parameters extracted from the LCP pump trace. **b**, Stokes parameters extracted from the RCP pump trace. The polarization state can be described by these four parameters: $S_0$ represents the intensity, $S_1$ and $S_2$ are associated with linear polarization along two sets of orthogonal axes, and $S_3$ is associated with circular polarization. **c**, Table collecting the Stokes parameters integrated in a 0.5 THz range around the main peaks in **a**, **b**. The $(S_3^*/S_0^*)^2$ quantity can be considered as an indication of the average amount of circular polarization, and it takes a value of 1 for ideal circularly polarized light.

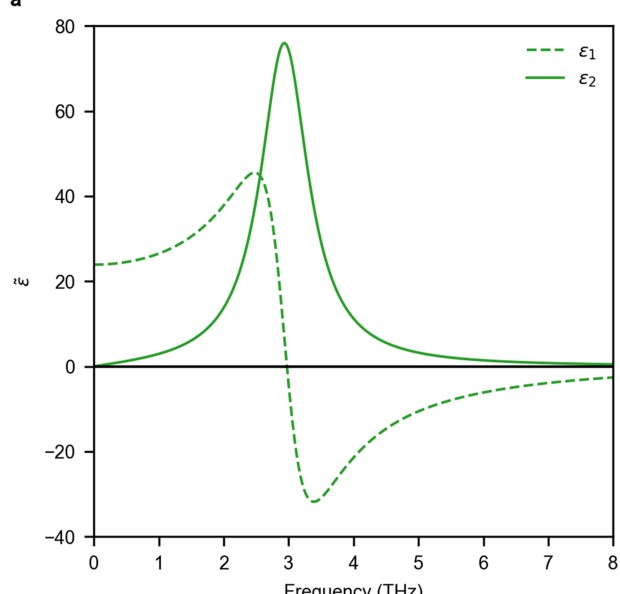

**a**

**b**

| T (K) | n | k | $l_{decay}$ (μm) |
|---|---|---|---|
| 160 | 1.03 | 4.42 | 3.60 |
| 180 | 1.20 | 4.72 | 3.37 |
| 200 | 1.49 | 5.07 | 3.14 |
| 220 | 1.77 | 5.38 | 2.96 |
| 240 | 2.18 | 5.73 | 2.78 |
| 260 | 2.50 | 5.94 | 2.68 |

| T (K) | n | k | $l_{decay}$ (μm) |
|---|---|---|---|
| 280 | 3.12 | 6.23 | 2.56 |
| 300 | 3.78 | 6.40 | 2.49 |
| 320 | 4.55 | 6.42 | 2.48 |
| 340 | 5.40 | 6.20 | 2.57 |
| 360 | 6.51 | 5.36 | 2.97 |
| 375 | 6.76 | 4.98 | 3.19 |

**Extended Data Fig. 4 | Complex permittivity and refractive index for STO in the terahertz regime. a**, Complex permittivity of STO at T = 300 K used for the simulations. The functional dependence was found assuming a Lorentz oscillator with linewidth and oscillator strength taken from ref. 55, while the resonance frequency was matched with the experimental soft phonon frequency of ref. 17 and listed in Extended Data Table 1. **b**, Temperature dependence of the real and imaginary parts of the refractive index in STO at 3 THz, and corresponding penetration depth $l_{decay}$. All values were estimated considering the dielectric function plotted in **a** and properly shifted to take into account the variation of the dielectric function with temperature.

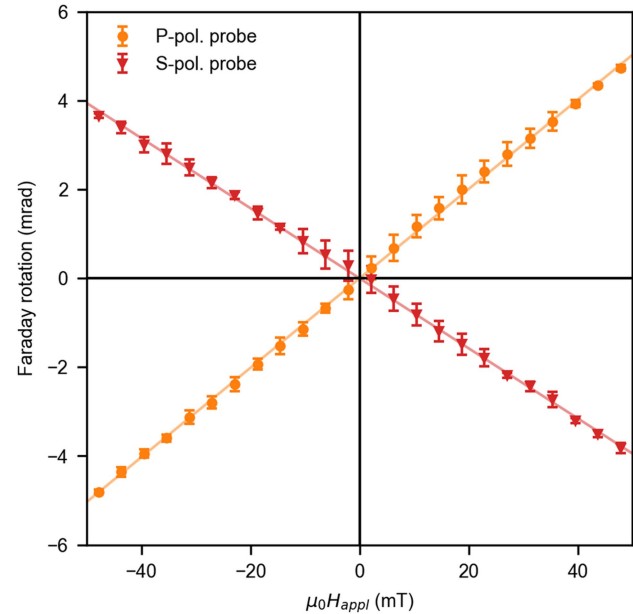

**Extended Data Fig. 5 | Magneto-optical Faraday effect in STO at 400 nm.**
Faraday rotation experienced by the transmitted 400 nm probe beam at
normal incidence when a magnetic field is applied in the direction of the probe
propagation (that is, perpendicularly to the sample surface). The measurements
were performed with a balanced detection scheme equipped with a half-wave
plate, and the P and S polarization directions were defined by the Wollaston
prism. After switching the probe polarization from P to S, the same balancing
orientation for the half-wave plate was maintained, in order to ensure consistency
between measurements. The solid lines represent linear fits to the data, which
allow us to estimate an average Verdet constant of approx. $180 \ rad \ m^{-1} \ T^{-1}$.
The change of sign when the polarization is switched from P to S is consistent
with a magnetic origin of the signal.

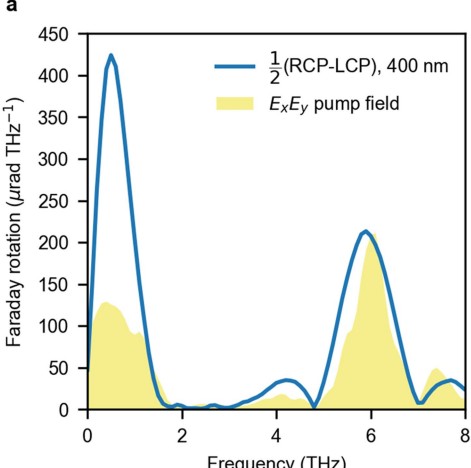

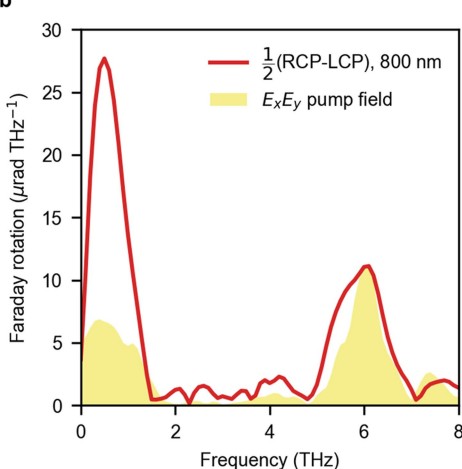

**Extended Data Fig. 6 | Measured polarization rotation in STO at 400 nm and 800 nm probe wavelengths. a**, Fourier transform of the measured Faraday rotation trace with a probe wavelength of 400 nm (solid blue curve), compared with the spectrum of the product of the two components $E_x$ and $E_y$ of the incident circular pump field (filled yellow area). **b**, Fourier transform of the measured Faraday rotation trace with a probe wavelength of 800 nm (solid red curve), corresponding to a photon energy of 1.55 eV, less than half of the direct bandgap in STO.

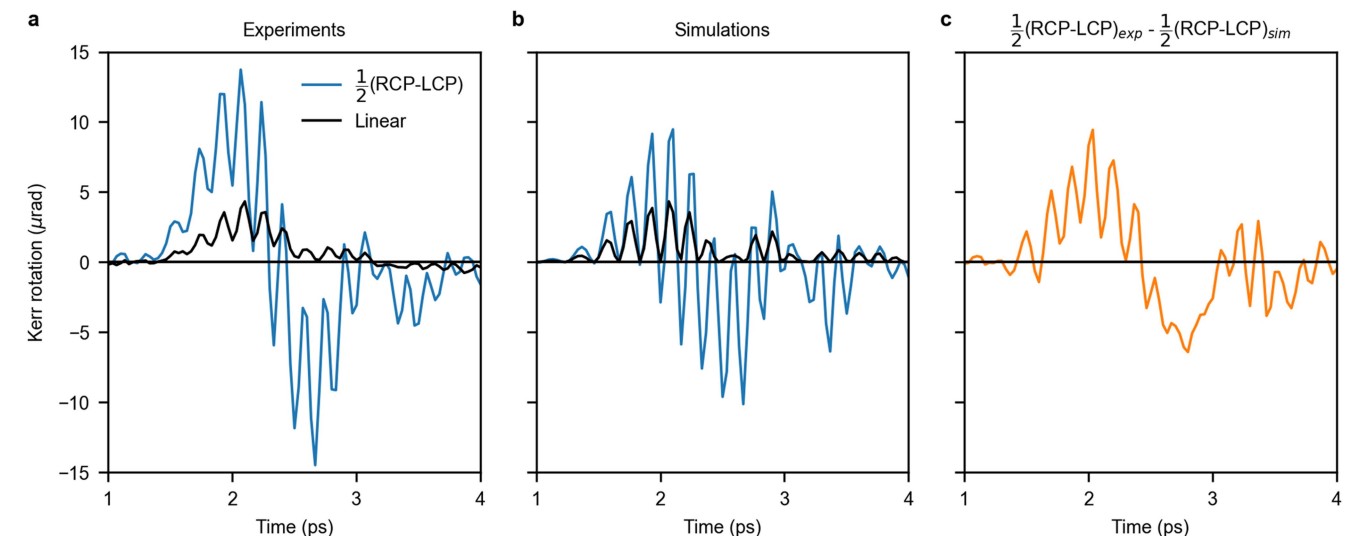

**Extended Data Fig. 7 | Response of STO to circularly and linearly polarized pump pulses. a**, Measured Kerr rotation signals for different pump helicities. The blue solid line shows the difference of the Kerr rotation signal between right (RCP) and left (LCP) circularly polarized terahertz fields. The black solid line is the Kerr rotation measured with a linearly polarized pump field of the same amplitude. **b**, Simulated Kerr rotation considering only the electronic Kerr effect (EKE) nonlinear contribution, as discussed in the Methods. Both the simulated signals were rescaled by the same factor in order to reproduce the measured amplitude of the experimental linear signal, for which the dynamical multiferroicity effect is not present. **c**, Difference between the experimental and simulated Kerr rotation for the circularly polarized pump case. See Methods for details.

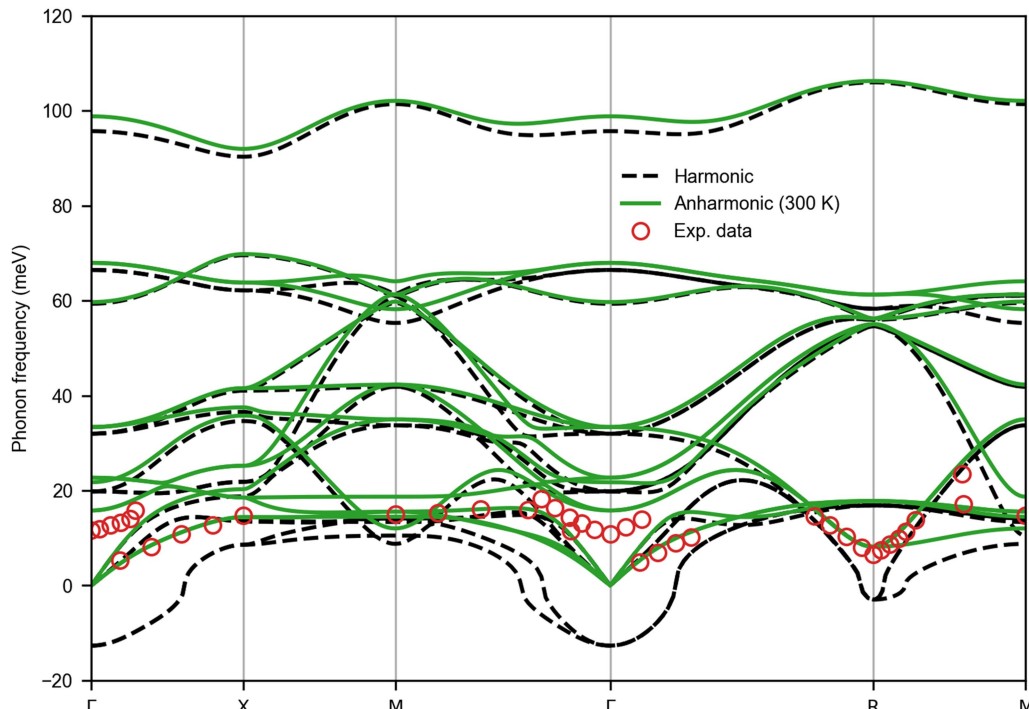

**Extended Data Fig. 8 | Phonon dispersion curves of cubic STO.** The curves were calculated based on the harmonic approximation (dashed lines) and the self-consistent phonon theory at 300 K (solid lines). The experimental inelastic neutron scattering data[18] are also shown for comparison (open symbols).

**Extended Data Table 1 | Phonon parameters used to solve the coupled oscillator equation**

| T (K) | $\omega_0/2\pi$ (THz) | $\Gamma/2\pi$ (THz) | $k$ (THz$^2$ Å$^{-2}$ AMU$^{-1}$) | $\chi$ (THz$^2$ Å$^{-2}$ AMU$^{-1}$) | $Z^*$ (e$^-$ AMU$^{-1/2}$) | $\gamma$ (e$^-$ AMU$^{-1}$) |
|---|---|---|---|---|---|---|
| 160 | 1.85 | 0.35 | 1556 | 127 | 1.62 | 0.057 |
| 180 | 2.00 | 0.41 | 1496 | 125 | 1.61 | 0.058 |
| 200 | 2.15 | 0.48 | 1436 | 123 | 1.60 | 0.059 |
| 220 | 2.27 | 0.52 | 1376 | 121 | 1.59 | 0.060 |
| 240 | 2.40 | 0.55 | 1312 | 119 | 1.58 | 0.061 |
| 260 | 2.48 | 0.58 | 1428 | 117 | 1.57 | 0.062 |
| 280 | 2.60 | 0.61 | 1180 | 115 | 1.56 | 0.062 |
| 300 | 2.70 | 0.63 | 1112 | 113 | 1.54 | 0.063 |
| 320 | 2.80 | 0.69 | 1044 | 110 | 1.52 | 0.064 |
| 340 | 2.90 | 0.74 | 976 | 108 | 1.50 | 0.064 |
| 360 | 3.05 | 0.77 | 908 | 105 | 1.49 | 0.065 |
| 375 | 3.10 | 0.81 | 840 | 102 | 1.47 | 0.065 |

The phonon centre frequency $\omega_0$ and linewidth $\Gamma$ are taken from the experimental values in ref. 17. The nonlinearity coefficient $k$, the nonlinear coupling coefficient $\chi$, the effective charge $Z^*$ and the gyromagnetic ratio $\gamma$ are taken from ab initio calculations.

# Extended Data Table 2 | Polarization vectors

| | | Harmonic | | | Anharmonic (T = 300 K) | | |
|---|---|---|---|---|---|---|---|
| | $k$ | $\tilde{e}^x(k)$ | $\tilde{e}^y(k)$ | $\tilde{e}^z(k)$ | $e^x(k)$ | $e^y(k)$ | $e^z(k)$ |
| Mode $Q_1$ | | | | | | | |
| | Sr | -0.191 | 0 | 0 | -0.415 | 0 | 0 |
| | Ti | -0.552 | 0 | 0 | -0.299 | 0 | 0 |
| | O1 | 0.511 | 0 | 0 | 0.483 | 0 | 0 |
| | O2 | 0.445 | 0 | 0 | 0.503 | 0 | 0 |
| | O3 | 0.445 | 0 | 0 | 0.503 | 0 | 0 |
| Mode $Q_2$ | | | | | | | |
| | Sr | 0 | -0.191 | 0 | 0 | -0.415 | 0 |
| | Ti | 0 | -0.552 | 0 | 0 | -0.299 | 0 |
| | O1 | 0 | 0.445 | 0 | 0 | 0.503 | 0 |
| | O2 | 0 | 0.511 | 0 | 0 | 0.483 | 0 |
| | O3 | 0 | 0.445 | 0 | 0 | 0.503 | 0 |
| Mode $Q_3$ | | | | | | | |
| | Sr | 0 | 0 | -0.191 | 0 | 0 | -0.415 |
| | Ti | 0 | 0 | -0.552 | 0 | 0 | -0.299 |
| | O1 | 0 | 0 | 0.445 | 0 | 0 | 0.503 |
| | O2 | 0 | 0 | 0.445 | 0 | 0 | 0.503 |
| | O3 | 0 | 0 | 0.511 | 0 | 0 | 0.483 |

The polarization vectors of the triply degenerate ferroelectric soft modes at $q$ = 0 calculated based on the harmonic approximation and the self-consistent phonon theory at 300 K. The modes $Q_1$, $Q_2$, and $Q_3$ correspond to the displacements in the [100], [010], and [001] directions, respectively. The fractional coordinates of each atom are as follows: Sr (0, 0, 0), Ti (1/2, 1/2, 1/2), O1 (0, 1/2, 1/2), O2 (1/2, 0, 1/2), O3 (1/2, 1/2, 0).