## [Peer Review File · Nature]

Manuscript Title: Terahertz electric field driven dynamical multiferroicity in SrTiO₃

Reviewer Comments & Author Rebuttals

Reviewer Reports on the Initial Version:

Referees' comments:

Referee #1 (Remarks to the Author):

The authors present a study of the vibrational response of paraelectric strontium titanate (SrTiO₃) when subjected to the irradiation with a circularly polarized terahertz pulse that is tuned into resonance with the infrared-active soft mode of the material. The authors measure a magnetic-optical Kerr response containing both difference- and sum-frequency components at 0.6 THz and 6 THz. They show that the difference-frequency signal can not be explained by an electronic $\chi^{(3)}$ effect alone, and suggest that an additional contribution from circularly driven phonons comes into play, which produce a magnetic moment according to the recently predicted dynamical multiferroic effect. The authors support their claim by temperature dependent measurements that show that the effect is resonant with the soft phonon frequency and they perform first-principles calculations that support the interpretation that a phonon magnetic moment qualitatively explains the missing response. They find that the magnetic moment of the phonon is orders of magnitude larger than originally predicted however, which indicates the presence of some sort of electron-phonon coupling.

In my opinion, the findings are intriguing, as they represent the first ever experimental demonstration of phonon-generated magnetism in nonmagnetic materials, which has the potential for great impact across condensed matter physics, materials science, and nonlinear optics communities. Both the experimental and computational data presented seem sound and the interpretations drawn from them about the physical origin of the mechanism are conclusive. I think that the manuscript is a good fit for the scope of Nature, but I have a number of questions and comments that I believe the authors should address, before proceeding towards publication. I will elaborate on these points below.

1) Page 2: The authors write that "The polar nature of the phonons implies a polarization P orthogonal to its time derivative $d_t P$ (...)." Shouldn't rather the components of P , P_x and P_y , be orthogonal to each other? The time derivative doesn't change the direction of the vector P , meaning P and $d_t P$ are collinear.

2) Fig. 2b: For perfectly circularly polarized light, the difference frequency component of the square of the electric field should peak at $\omega_- = 0$. When looking at the Extended Data Figure 1b, it seems like the major peak at 0.6 THz possibly arises from the frequency mixing of the E_x peak with the left E_y shoulder.

3.1) Page 5: The authors write that they rule out an influence of the inverse Faraday effect because of their low pump photon energy compared to previous studies. The inverse Faraday effect should be present at any frequency range however, even in the terahertz regime, see e.g. [Phys. Rev. B 84, 174433 (2011)] for a metal, and should in principle be included in the simulations leading to Figs. 4a and b. How can these two electronic effects ($\chi^{(3)}$ and IFE) be distinguished here without knowing the order of magnitude of the inverse Faraday effect?

3.2) Connecting to the above point: It was recently proposed that the inverse Faraday effect does not produce a sum-frequency component for circularly polarized light, [Phys. Rev. B 103, 094407

(2021)], which means that it would only contribute to the difference-frequency Kerr signal here. When including it in the simulations for Fig. 4b, it could potentially fill up the difference-frequency peak a bit and the remaining phononic effect would have to be extracted from $\text{total_signal} - [\text{signal}(\chi^3) + \text{signal}(\text{IFE})]$. The big question here however is the magnitude of the IFE - it is probably small for nonmagnetic materials for excitations well below the band gap, but I don't think that one can rule it out per se. The inverse Faraday effect can still be distinguished from the phononic effect through tuning the laser in and out of resonance with the phonons, as outlined in another recent study [Phys. Rev. Research 4, 013129 (2022)].

4) Page 5: The authors outline a number of recent studies that find effective phonon magnetic moments that are much larger than those theoretically predicted. Another more recent one is [arXiv:2208.12235]. There have also been theoretical studies that predict large effects, e.g. [Phys. Rev. Research 2, 043035 (2020)] and [Phys. Rev. Research 4, 013129 (2022)]. A particularly relevant study in the context of the present manuscript however is [arXiv:2208.05746], where the authors predict an effective magnetic moment of 0.2 Bohr magneton in electron-doped KTaO₃ arising from different couplings to the electronic spin-up and spin-down channels of the nonmagnetic material. KTaO₃ has many characteristics similar to SrTiO₃ and the authors should comment on whether this proposal could be a candidate for explaining the magnitude of the effect measured here.

5) Fig. 4: The authors interpret the originally described dynamical multiferroic effect as a phonon manifestation of the Barnett effect, where mechanical rotation is translated to magnetization. The Barnett effect conventionally describes the translation of macroscopic rotation of a solid body to magnetization - the phonons here would be an intermediate step, see Ref. 36 in the manuscript. Another recent interpretation considers the phenomenon a phonon manifestation of the inverse Faraday effect, [Phys. Rev. Research 2, 043035 (2020)], which could be set in context here.

6) General: Is there an estimate for how much energy gets deposited in the material by the laser pulse (e.g. in eV/unit cell, both in the phonon and electron systems) and by how much the temperature of the material increases due to that? Does this have an impact on the temperature-dependent measurements of the Kerr response?

7) Methods, phenomenological model: The authors use an effective electric field \tilde{E} in their calculations that represents the amount of field that actually penetrates the sample. Where does the conversion factor α come from here?

Minor comment

- Page 7: Schlauderer et al. reference in text leads to Ref. 38, not Ref. 39.

Referee #2 (Remarks to the Author):

Dear Tobias,

In the manuscript, the authors Basini et al., presented an experimental demonstration of dynamical multiferroicity in SrTiO₃. A theoretical concept of the same was proposed by Juraschek et al. back in 2017. This work demonstrates that phenomenon and as highlighted by both works, the evidence bears practical relevance and is a significant development in the field. By using intense circularly-polarized THz light, the authors induced multiferroicity in an otherwise boring paraelectric and diamagnetic SrTiO₃ in rather a dynamic manner. The authors supported their results both qualitatively using ab-initio calculations and quantitatively using the Barnett effect. There are, however, several points that to my opinion are unclear or the description is quite terse. I could potentially recommend the manuscript for publication if my concerns (as listed below) are satisfactorily answered.

Below I summarize my concerns:

1. On the experimental side, it would be important to know how the two Kerr rotation scans for opposite helicities were taken, where the authors show the two circularly polarized THz pulses with opposite helicities first. Then, the corresponding Kerr signal for the two helicities and then finally the difference Kerr signal. This would facilitate the understanding of the experimental sequence better (for non-specialists). In addition, the demonstration of opposite helicities will add more concreteness in the obtained results along with the methodology undertaken to calibrate the circular THz electric field values.
2. The description on the comparison of two peaks in Fig. 2b is quite terse. Compared to the dynamical multiferroicity model predicted by Juraschek et al., the amplitude of the high frequency peak is significant as observed experimentally. While the authors associate this to the electronic nonlinearities, they do not explicitly mention the microscopy of this electronic nonlinearity. They mention its associated with third-order susceptibility, but this does not clarify at all.
3. An apparent question that comes to my mind: What happens to the difference Kerr signal as the mode systematically softens to 0.2 THz at 5K. What happens to the ratio of the a'' and a' ?
4. It is not clear to me why the authors state that the quadratic dependence of a' mode on the driving THz field is an indication of extra contribution and how is it related to electronic nonlinearity?
5. Lines 205-206, has very little grounding, being a speculation. I am not sure if I completely understand what the authors are trying to imply here. How does the transfer of angular momentum to valence electrons leads to a larger enhancement and why only the high frequency mode? I fail to find to good train of thoughts that connects it all. A careful rephrasing and proper justification (instead of speculation) would be crucial for the said claim.
6. How does the overall result apply to materials other than STO? The conclusion is too much focused around STO. In addition to the proposed possible implications of their results, the authors need to bring out the generality of the obtained results.
7. Minor one: In the figure caption of Extended Data Fig. 2, the authors mentioned about oscillator strength found in Ref. 44. I think it is typo, I did not find any such expressions in Ref. 44, did they mean Ref. 43 instead?

Referee #3 (Remarks to the Author):

Summary of key results:

Basini et al. report the observation of an optical polarization rotation in SrTiO₃ driven by circularly polarized THz light pulses that exhibits some interesting features. Most notably, the pump-helicity dependent component of the polarization rotation has frequency content at roughly zero frequency and twice the driving frequency (indicating a response quadratic in the driving field), and the magnitude of the response shows a non-monotonic temperature dependence (peaking close to the temperature where the THz pump is in resonance with the soft polar phonon of SrTiO₃). Based on comparisons to the theory of “dynamical multiferroicity,” the authors attribute part of this signal to a magnetic moment arising due to circulating ionic motions from the resonant phonon excitation (with the other part being explained by a “trivial” nonlinear optical response). Remarkably, the induced moment deduced from these measurements is found to be four orders of magnitude larger than theoretical predictions.

Originality and significance:

This work contributes to the growing field of ultrafast optical control of materials in general, and of “nonlinear phononics” (in which resonant vibrational control is leveraged to create non-equilibrium structures and phases), in particular. The idea of “dynamical multiferroicity” upon which this experiment is based was proposed in 2017. It essentially rests on the phenomenological notion that the time-varying polarization associated with charged ions moving in closed loops acts like a magnetic field and can generate an internal magnetization, even in an otherwise non-magnetic material. The effect arises naturally from Maxwell’s equations (it’s basically a statement of Ampere’s Law). It can also be considered the ionic equivalent of the inverse Faraday effect, which is more commonly realized at optical frequencies (as pointed out by the authors in the main text), but the difference in frequency does not preclude the two phenomena from having the same origin. From that standpoint, I don’t even see the underlying phenomenon that the experiment seeks to demonstrate particularly novel or as providing a meaningful advance in our understanding.

On the other hand, from a practical point of view, it is interesting to explore new ways to control materials’ properties on ultrafast timescales using light. However, the principle of resonantly driving circular atomic motions to create effective magnetic fields was already experimentally demonstrated by Nova et al. in 2016 (Ref. 21). The new aspect of the work by Basini et al. is to apply this technique at low frequencies in a non-magnetic material. These aspects don’t provide any new insights in my view. It is somewhat intriguing that they observe such a large induced magnetic moment in their experiment, but I have my doubts about the technical aspects leading to their conclusions (see below). Even putting those doubts aside, though, based on the considerations above, both in terms of the theoretical framework and experimental validation, I do not consider the results presented in this paper to be highly original or significant.

Data, methodology, conclusions

The most important experimental conclusions the authors extract from their data are that (a) there is a magnetic contribution to their signal which arises from the dynamical multiferroic effect, and that (b) this magnetic contribution is much larger than expected indicating some additional mechanism at play. Making such statements from the optical data is highly non-trivial, and upon thoroughly examining the experimental data and methodologies presented in the main text and supplement, I do not think there is sufficient experimental evidence to support their claims. Below I detail some of my technical concerns:

- Is the THz excitation really circular? Is the helicity dependence accurately measured?

The authors use a bandpass filter and a quartz birefringent waveplate to create circularly polarized pulses centered at 3 THz. How confident are the authors that the output of the waveplate really produces circular light? Even at IR and visible wavelengths where optics are typically of much higher quality and better characterized, slight misalignments or non-ideal optical elements can lead to differences in the polarization state. Indeed, in the supplement, the authors show EOS traces of the E_x and E_y components which have obviously different FFT spectra, which indicates that the THz light cannot be circular (would rather be somewhat elliptical). The authors also do not mention how they measure the E_x vs. E_y components nor do they mention how they obtain the opposite THz helicities. Presumably the opposite helicity is achieved by rotating their quarter waveplate by 90 degrees and remeasuring (please correct if I'm wrong, more details would be appreciated). Again, even if the waveplate was ideal, it is unlikely that this rotation can be done with perfect precision. These non-idealities will lead to a number of artifacts, one of which is some non-canceling linear components of the signal such that the polarization rotation they measure does not reflect a "MOKE" signal.

- How is the field strength calibrated?

The authors state in the supplement "The exact size of the beam is not crucial to estimate the fluence since we characterised the electric field of the radiation." If the EOS is properly calibrated, this can certainly be true. But, then when discussing the EOS in the methods "In panel (a) the reflected terahertz field from the back side of the GaP crystal starts to interfere with the direct beam at approximately $t = 3.8$ ps. This complicates the electro-optic sampling data presented here..." So, with these apparent artifacts in the EOS data it is unclear to me how the authors actually calibrated the field strength in their experiment. Also, the correspondence between the field measured by EOS and the field incident on the sample requires that the EOS crystal be placed in the exact same position as the sample. Was this done and to what accuracy? There are no error bars anywhere in this paper, which to me is a huge huge red flag.

- What about other contributions to the signal? (Related to previous questions)

One of the most important the authors make in trying to interpret their data as an induced magnetic moment is that when comparing the measured pump-induced polarization rotation to the product of the E_x and E_y fields, there is a difference in the low-frequency peak of the spectrum. That is, if one

assumes that there is some “trivial” third-order nonlinear optical effect that can also create a polarization rotation (i.e. pump-induced birefringence), then the authors say that this effect should be proportional to $E_x E_y$, and any part of the signal which is not strictly proportional to this product is magnetic. I have many problems with this line of reasoning.

First of all, the authors do not even provide a cursory argument in the paper for why the third-order optical effect should go like $E_x E_y$ nor do they account for the strong polarization dependence expected from this response, simply referencing a separate study. The problem is that this separate study doesn't exist yet! To me, it is totally unacceptable to defer a key aspect of the data analysis to a paper which is still in preparation. For argument's sake, let's assume the authors assumptions are valid, then the calibration and measurement of the exact fields E_x and E_y becomes extremely crucial, but based on the data and method details presented, the accuracy of these fields are highly questionable.

Secondly, the authors do not try to rule out any other possible contributions to the low-frequency peak before attributing it to magnetism. For example, in the text they point out “intense coherent THz excitation of the soft mode in STO can lead to highly nonlinear phonon responses that overcome the quantum fluctuations creating a ferroelectric order absent at equilibrium” (Ref. 38). There was also other work that showed long-lived ferroelectric states upon phonon pumping (Nova et al. Science, 2019 <https://doi.org/10.1126/science.aaw4911>) which should be cited. The experimental work here uses intense coherent THz excitation, so we might expect ferroelectric order to develop in the same way as these works. Obviously, such a phase transition could drastically change the optical properties in a highly nonlinear way, including the reflectivity/refractive index at the probe frequency, which could lead to spurious low frequency content in the polarization rotation (I will also come back to this point later). There are other possible artifacts that must be ruled out as well, including polaritonic effects, higher order nonlinearities, etc.

One measurement which would be very valuable in this context is probe polarization dependence, which should exhibit very different behavior if the signal arose from magnetism vs. alternative optical effects. Has this been done?

- How is the magnetic moment obtained?

Again, one of the key claims is that the magnetic moment observed is four orders of magnitude larger than what would be expected from theory. There are several aspects of this conversion that need clarification.

First, are the authors considering only the “extra” magnetic part of the low frequency signal when extracting the magnetic moment? If so, how can this reliably be done without having the proportionality factor relating $E_x E_y$ to the nonlinear optical background that is supposed to be removed?

Second, the authors take the literature value for the Verdet constant from transmission measurements and use it as their MOKE calibration. This is not a sound way to obtain the magnetic moment from MOKE data. The authors note that they measured also in transmission and only saw differences up to a factor of two, but this does not mean that the literature Faraday coefficient can be used. The Verdet constant (both magnitude and sign) will be highly sensitive to wavelength near the band edge where the

probe frequency is parked in this experiment, for example. So, one really needs in the best case scenario to directly measure the static MOKE signal as a function of field to have an accurate calibration on the actual sample using the real experimental optical parameters. Barring that, one at least needs to also take into account changes in reflectivity and in the magneto-optical coefficients that might also be driven by the THz field. As mentioned above, these effects can be significant since it's known that STO is highly polarizable by the THz field and can even undergo a phase transition, which would drastically change the optical spectra.

Thirdly, the extracted moment is highly dependent on the estimated pump-probe penetration depth mismatch, which may also exhibit nonlinear and time-dependent behavior upon pumping.

- Other issues

If I assume that the effect is driven by a phonon-induced magnetization, I would expect from the equation $M \sim P \times dP/dt$ that the signal should have a rectified contribution and a contribution at twice the phonon frequency. But, in the temperature dependence of the signal, a peak is always found in the spectrum at twice the THz frequency (6 THz) even though the phonon frequency is changing a significant amount over that range. Why are features not seen at twice the phonon frequency instead?

The calculated moment time dependence Fig. 4a only shows a single dip, while the actual signal clearly shows a bipolar behavior where it has a positive rectified component around 0.5 ps and the negative part around 1-1.5 ps. What is the origin of this "up-down" behavior and why is not reproduced by the theoretical calculation?

Even if I take the arguments in the paper at face value, and I assume that the ionic motions really are creating an effective B-field, I have trouble understanding where the resulting magnetization comes from. SrTiO₃ is a d⁰ compound. There are no electrons in the d-shell which can be polarized by the effective field. If the idea is that angular momentum is being transferred from the phonons to the electronic system, can the authors explain what electronic system is being coupled to?

The raw data, in addition to the helicity dependent data should be given somewhere, so readers can assess the data analysis better. It would also be helpful to have the pump e-field superimposed on top of the rotation signal.

The fact that the signal gets largest when the phonon is in resonance with the 3 THz pulse is not necessarily evidence that the signal comes from this dynamical multiferroicity – even a "trivial" nonlinear optical effect will see a resonant enhancement of this type.

Final comments:

To reiterate: this paper presents an attractive picture of using circularly polarized light to create magnetization in a non-magnetic material by driving “rotating” phonons. On the technical side, the evidence for this picture is highly circumstantial, and I am not convinced that the analysis is sufficiently comprehensive and that all relevant artifacts have been ruled out. More generally, in the context of the field of optical control/nonlinear phononics, I do not think the results of this paper constitute a highly original or significant contribution. While I do think the work presented here is valuable in a more narrow context and should eventually be published somewhere, in my opinion, it is not appropriate for Nature, and it would certainly need more comprehensive data analysis and some additional experiments as presented above before publication anywhere.

Author Rebuttals to Initial Comments:

Rebuttal letter

We would like to thank all three referees for their careful work of reviewing our manuscript, which helped us to improve the revised version. We already had available some of the data required by the referees, but we also had to perform additional measurements to answer all of their questions. There has been an issue with the main laser system, hence the delayed response. We think that we have now addressed all the concerns and we hope that all referees will recommend our work for publication

A general statement that we would like to make, before going into each individual point, is the following. There are suggestions in some of the remarks by the referees on whether the effect that we observe could be explained in terms of the inverse Faraday effect. We will go in detail into those when answering the referees' comments point by point; here, we want to stress both a semantic and a conceptual difference between the inverse Faraday effect and the effect that we propose. It took us some time as well to clearly grasp this nuance, so we understand where the suggestions from the referees come from. The point is that the inverse Faraday effect is a *phenomenological* description of any light-induced effect with a given symmetry. We completely agree with the fact that our proposed effect has the very same symmetry of the inverse Faraday effect, as already pointed out in the original work predicting the dynamical multiferroicity. However, and this is the crucial difference, *dynamical multiferroicity is a microscopic mechanism*, describing, in our case, the resonant behavior with the soft phonon. To put it differently, the mechanism for the inverse Faraday effect requires the light field to transduce the angular momentum into the orbital motion that results in magnetization. In our case, the transduction occurs between the angular momentum of lattice modes, and the magnetic moment of the electrons. *The light drive is secondary, as one could in principle induce the same effect using chiral phonons excited without a light source.* We can say that dynamical multiferroicity is an effect belonging to the generic symmetry class of “inverse Faraday effects”, as there are many of those. They are of various microscopic origins, depending on the wavelength of the light used, and of the characteristics of the materials (e.g., metallic vs insulating, magnetic vs non-magnetic, etc.). But we argue that one cannot put dynamical multiferroicity and inverse Faraday effect (which is also usually assumed to be an electronic effect, not a phononic one) in opposition to each other, without specifying. It is like comparing the content with the container: there can not be any logical dichotomy between them. We hope that our revised manuscript, and our rebuttal, will help clarify this issue for potential readers.

We note that an extremely recent manuscript that has appeared on the arXiv [D. C. S. Davies, et al. Phononic Switching of Magnetization by the Ultrafast Barnett Effect, arXiv:2305.11551 (2023)] while finalizing this rebuttal, shows that our suggested mechanism is correct, and one of its applications that we proposed in the original manuscript does work: the overall magnetic moment induced by circular phonons in a crystalline substrate can be of the order of the electronic and not the phononic one (amplified by the Barnett effect), and become so large to switch the magnetization of a thin film ferromagnet grown on top of that substrate. In that work, the authors were not able to study the details of the mechanism in the time-domain in the way we did, but their results confirm our original hypothesis. We still reply to all issues raised by the referees point by point here below, but we hope all the referees will now be convinced by independent data validating our prediction.

Referee #1 (Remarks to the Author):

The authors present a study of the vibrational response of paraelectric strontium titanate (SrTiO₃) when subjected to the irradiation with a circularly polarized terahertz pulse that is tuned into resonance with the infrared-active soft mode of the material. The authors measure a magnetic-optical Kerr response containing both difference- and sum-frequency components at 0.6 THz and 6 THz. They show that the difference-frequency signal can not be explained by an electronic $\chi^{(3)}$ effect alone, and suggest that an additional contribution from circularly driven phonons comes into play, which produce a magnetic moment according to the recently predicted dynamical multiferroic effect. The authors support their claim by temperature dependent measurements that show that the effect is resonant with the soft phonon frequency and they perform first-principles calculations that support the interpretation that a phonon magnetic moment qualitatively explains the missing response. They find that the magnetic moment of the phonon is orders of magnitude larger than originally predicted however, which indicates the presence of some sort of electron-phonon coupling.

In my opinion, the findings are intriguing, as they represent the first ever experimental demonstration of phonon-generated magnetism in nonmagnetic materials, which has the potential for great impact across condensed matter physics, materials science, and nonlinear optics communities. Both the experimental and computational data presented seem sound and the interpretations drawn from them about the physical origin of the mechanism are conclusive. I think that the manuscript is a good fit for the scope of Nature, but I have a number of questions and comments that I believe the authors should address, before proceeding towards publication. I will elaborate on these points below.

We thank the referee for the very positive evaluation about our work and for recognizing its novelty. We believe that we have addressed all of the issues raised, and we hope he or she will recommend the revised version for publication.

1) Page 2: The authors write that "The polar nature of the phonons implies a polarization P orthogonal to its time derivative $d_t P$ (...)." Shouldn't rather the components of P , P_x and P_y , be orthogonal to each other? The time derivative doesn't change the direction of the vector P , meaning P and $d_t P$ are collinear. We thank the referee for noticing this issue. It is indeed the *circular* nature of the polar phonon that implies a polarization P orthogonal to its time derivative $d_t P$. We corrected the text accordingly.

2) Fig. 2b: For perfectly circularly polarized light, the difference frequency component of the square of the electric field should peak at $\omega_- = 0$. When looking at the Extended Data Figure 1b, it seems like the major peak at 0.6 THz possibly arises from the frequency mixing of the E_x peak with the left E_y shoulder.

We thank the referee for this comment. To clarify this concept, please note that Fig. 4b shows the amplitude spectrum of the induced magnetic moment (orange curve) generated by the term $\vec{P} \times \partial_t \vec{P}$, which is peaked at $\omega = 0$, as predicted within the framework of dynamical multiferroicity. On the other hand, two peaks of comparable amplitude can be obtained when accounting for the electronic nonlinear response associated with $\chi^{(3)}$, as highlighted in the draft. Indeed, a term proportional to $E_x \cdot E_y$ (pink shaded area in Fig. 4b), combined with a non-perfect circularly polarized light produced by the THz-grade

crystal quarter-wave plate, give rise to a low-frequency peak, whose maximum is reached for $\omega \neq 0$. The total response, which combines the magnetic and the $\chi^{(3)}$ responses, still shows a peak at $\omega \neq 0$, whose area is enhanced thanks to the magnetic contribution with respect to the bare $\chi^{(3)}$ response. Details are given in Ref. [M. Basini, M. Udina et al. arXiv:2210.14053] which was unfortunately and unexpectedly made available only a few weeks after the submission of the original manuscript.

3.1) Page 5: The authors write that they rule out an influence of the inverse Faraday effect because of their low pump photon energy compared to previous studies. The inverse Faraday effect should be present at any frequency range however, even in the terahertz regime, see e.g. [Phys. Rev. B 84, 174433 (2011)] for a metal, and should in principle be included in the simulations leading to Figs. 4a and b. How can these two electronic effects ($\chi^{(3)}$ and IFE) be distinguished here without knowing the order of magnitude of the inverse Faraday effect?

We thank the referee for this important observation. We point back to the very first remark of this rebuttal. We agree that the effect measured in the suggested reference has indeed the same symmetry of an effect belonging to the IFEs class, but it is of very different microscopic origin. In [Phys. Rev. B 84, 174433 (2011)], the IFE generated by a circularly polarized electric field of 100kV/cm amplitude and 1 THz frequency was evaluated in a metal (Pt), with spin-orbit coupling $\lambda E_F^2=0.5$, $\lambda = 2\pi h/(4m^2 c^2)$, $E_F = 1$ eV [Sci. Technol. Adv. Mater. 9 (2008) 014105]. In this system, the effective magnetic field generated by IFE is described in terms of the response of the conduction electrons, not present in our insulating material. Furthermore, the calculated value is $B_{\text{eff}}=10^{-4}$ T in Pt, two orders of magnitude lower than the $B_{\text{eff}} = 0.032$ T measured in our work. Using the formula of Ref. [Phys. Rev. B 84, 174433 (2011)] we would, on the contrary, expect an even smaller effect, due to the lack of available conduction electrons. Hence, these estimates confirm that we are indeed looking at a different mechanism.

3.2) Connecting to the above point: It was recently proposed that the inverse Faraday effect does not produce a sum-frequency component for circularly polarized light, [Phys. Rev. B 103, 094407 (2021)], which means that it would only contribute to the difference-frequency Kerr signal here. When including it in the simulations for Fig. 4b, it could potentially fill up the difference-frequency peak a bit and the remaining phononic effect would have to be extracted from $\text{total_signal} - [\text{signal}(\chi^{(3)}) + \text{signal}(\text{IFE})]$. The big question here however is the magnitude of the IFE - it is probably small for nonmagnetic materials for excitations well below the band gap, but I don't think that one can rule it out per se. The inverse Faraday effect can still be distinguished from the phononic effect through tuning the laser in and out of resonance with the phonons, as outlined in another recent study [Phys. Rev. Research 4, 013129 (2022)].

We thank the referee for the observation. We believe that the rebuttal to the previous issue answers this one as well, regarding the magnitude of the effect. We also point out that our temperature-dependent measurements are a way to tune the phonon frequency. This is analogous to tuning the terahertz laser frequency (which is practically more complicated to implement in the lab), and demonstrates the fact that what we observe is a phononic effect, not an electronic one.

4) Page 5: The authors outline a number of recent studies that find effective phonon magnetic moments that are much larger than those theoretically predicted. Another more recent one is [arXiv:2208.12235].

There have also been theoretical studies that predict large effects, e.g. [Phys. Rev. Research 2, 043035 (2020)] and [Phys. Rev. Research 4, 013129 (2022)]. A particularly relevant study in the context of the present manuscript however is [arXiv:2208.05746], where the authors predict an effective magnetic moment of 0.2 Bohr magneton in electron-doped KTaO₃ arising from different couplings to the electronic spin-up and spin-down channels of the nonmagnetic material. KTaO₃ has many characteristics similar to SrTiO₃ and the authors should comment on whether this proposal could be a candidate for explaining the magnitude of the effect measured here.

We thank the referee for the suggestion, we now included [arXiv:2208.12235], [Phys. Rev. Research 2, 043035 (2020)] and [Phys. Rev. Research 4, 013129 (2022)] in the list of references. Moreover, even if we agree on the general relevance of the mechanism proposed in [arXiv:2208.05746], we ruled it out for our system. Indeed, in order to produce an appreciable magnetization in KTaO₃, the calculations in Ref. [arXiv:2208.05746] show that the position of the Fermi level should be very close to, or just above the conduction band. This is clearly not the case of the undoped bulk SrTiO₃ sample that we investigated in our work, and we can rule that out. A relatively large doping or sizable oxygen vacancies would be required [Physical Review B 83, 214107 (2011)].

5) Fig. 4: The authors interpret the originally described dynamical multiferroic effect as a phonon manifestation of the Barnett effect, where mechanical rotation is translated to magnetization. The Barnett effect conventionally describes the translation of macroscopic rotation of a solid body to magnetization - the phonons here would be an intermediate step, see Ref. 36 in the manuscript. Another recent interpretation considers the phenomenon a phonon manifestation of the inverse Faraday effect, [Phys. Rev. Research 2, 043035 (2020)], which could be set in context here.

We thank the referee for the remark about the Barnett effect being defined as a mechanical rotation. This is indeed correct, but we kept the same name in analogy with the choice of Ref. 35 in the original manuscript, where the phonon version of the Einstein – de Haas effect was given the same name as the original effect. We believe that this is an acceptable approximation, and we slightly changed the text of the revised manuscript to emphasize even more the fact that here we refer to a transient phononic response. As for the second comment, we point again back to the IFE discussion at the beginning of this rebuttal.

6) General: Is there an estimate for how much energy gets deposited in the material by the laser pulse (e.g. in eV/unit cell, both in the phonon and electron systems) and by how much the temperature of the material increases due to that? Does this have an impact on the temperature-dependent measurements of the Kerr response?

We thank the referee for this question, and we agree it is a relevant one. In the original manuscript, we had estimated the pump fluence to be 60 $\mu\text{J}/\text{cm}^2$ (see Methods). With this value, we can compute the absorbed energy density knowing the material parameters. At 300 K, the penetration depth of the pump electric field is approximately 2.5 μm and the absorptance is 0.24, which leads to an estimate of the average energy density absorbed by the sample of $115.7 \text{ mJ}/\text{cm}^3 = 0.043 \text{ meV}/\text{unit cell}$. The temperature increase determined by such an energy density can be evaluated by knowing the STO heat capacity and density. Considering a density of 5.18 g/cm^3 [MTI Corporation], a heat capacity of 100 $\text{J}/(\text{K mol})$ [D. de Ligny and P. Richet, Phys. Rev. B 53, 3013 (1996)], and a molar mass of 183.5 g/mol , the temperature increase is expected to be 0.04 K, a value that can be neglected during the temperature-dependent measurements presented in the draft. We have now added these numbers in the revised Methods. An

independent experimental confirmation of negligible heating is found in Ref. [Appl. Phys. Lett. 110, 081106 (2017)]. There, time-resolved hard x-ray diffraction showed that even for terahertz fields as high as 900 kV/cm (four times larger than ours, hence with a pulse energy sixteen times larger), no measurable thermal expansion was detectable in signal from the Bragg peak. The first indication of temperature-induced changes to the lattice was observed only when a metamaterial was fabricated on top of the STO, leading to an additional five-fold enhancement of the electric field, or a 25 times increase of the intensity. Hence, heat effects require 20 times larger electric fields (400 times larger energy density) to cause a measurable temperature change of the order of 10s of Kelvin. We can hence safely neglect them in our measurements.

7) Methods, phenomenological model: The authors use an effective electric field \tilde{E} in their calculations that represents the amount of field that actually penetrates the sample. Where does the conversion factor α come from here?

We thank the referee for this observation. The local-field factor α is a proportionality constant relating the intensity of the incident electric field with the effective electric field inside the sample, to be used in the phenomenological model presented in the “Methods” section. In our work, we selected an α value of 0.7 being driven by the experiments, since we considered α as the only free parameter to be adjusted to reproduce the experimental results.

Our choice can be validated by looking for a rigorous definition of the appropriate local-field factor, which actually is a long-standing problem in literature, as described, e.g., by A. Aubret et al., Chem. Phys. Chem 20, 345 (2019). The exact formulation depends on the material of interest and on its symmetry, but for dielectrics two models are typically used: the Lorentz model, and the Onsager-Böttcher local-field approach, see P. de Vries and A. Lagendijk, Phys. Rev. Lett. 81, 1381 (1998). Briefly, the former model considers a “virtual” cavity inside the dielectric material (leading to the Clausius-Mossotti relation). The latter takes into account a “real” empty cavity inside the material, where the atom/molecule of interest resides. For $n = 3.8 + 6.4i$ (refractive index of STO at 3 THz and 300 K), the Lorentz local-field factor corresponds to $\alpha = 4.5$, and the Onsager-Böttcher local-field factor evaluates to $\alpha = 0.38$. Those two values then define the range for a realistic value of α , which is indeed compatible with the chosen $\alpha = 0.7$. We stress again that this is the only free parameter used in the calculations, all others are either from experiments or ab-initio. Finally, we note that the calculations of the local-field factor are particularly difficult for a material like STO when driven by a strong terahertz field. In fact, the action of the field on the ions changes their polarization, hence the local-field as well, which would need to be reevaluated in a self-consistent manner. This is beyond the scope of our work.

Minor comment

- Page 7: Schlauderer et al. reference in text leads to Ref. 38, not Ref. 39.

We thank the referee for noting the typo, We corrected the reference number.

Referee #2 (Remarks to the Author):

Dear Tobias,

In the manuscript, the authors Basini et al., presented an experimental demonstration of dynamical multiferroicity in SrTiO₃. A theoretical concept of the same was proposed by Juraschek et al. back in 2017. This work demonstrates that phenomenon and as highlighted by both works, the evidence bears practical relevance and is a significant development in the field. By using intense circularly-polarized THz light, the authors induced multiferroicity in an otherwise boring paraelectric and diamagnetic SrTiO₃ in rather a dynamic manner. The authors supported their results both qualitatively using ab-initio calculations and quantitatively using the Barnett effect. There are, however, several points that to my opinion are unclear or the description is quite terse. I could potentially recommend the manuscript for publication if my concerns (as listed below) are satisfactorily answered.

Below I summarize my concerns:

We thank the referee for summarizing our work and for finding it potentially suitable for Nature. We address in detail all the concerns raised by the referee below.

1. On the experimental side, it would be important to know how the two Kerr rotation scans for opposite helicities were taken, where the authors show the two circularly polarized THz pulses with opposite helicities first. Then, the corresponding Kerr signal for the two helicities and then finally the difference Kerr signal. This would facilitate the understanding of the experimental sequence better (for non-specialists). In addition, the demonstration of opposite helicities will add more concreteness in the obtained results along with the methodology undertaken to calibrate the circular THz electric field values.

We agree with the comment of the referee, and we thank him or her for giving us the opportunity to clarify this point. The field strength was calculated, using a standard procedure, first by measuring the time-resolved birefringence, at 800 nm wavelength, caused by the THz electric field in a 50 μm thick GaP crystal cut along the 110 direction. In order to do that, we used a balanced detection scheme with two diodes measuring the signals I_1 and I_2 produced by a quarter-wave plate and a Wollaston prism placed after the sample. Once the time trace was retrieved, we moved the delay stage to the time-delay where the maximum of the terahertz electric field was found. In the temporal axis of the Figure below, this point was 2.6 ps for the E_x component of the right circularly polarized RCP. Following Ref. [M. Hoffman *et al.* Optics Letters Vol. 36, Issue 23, pp. 4473-4475 (2011)], we computed $E_{\text{THz}} = \lambda_0 \sin^{-1}((I_1 - I_2)/(I_1 + I_2)) / (2\pi n_0^3 r_{41} t_{\text{GaP}} L)$, where $n_0 = 3.193$ is the refractive index of GaP at 800 nm, $L = 0.05\text{mm}$ the thickness of the crystal, $\lambda_0 = 800\text{ nm}$ the probe wavelength, $r_{41} = 0.88\text{ pm/V}$ [J. Phys. D: Appl. Phys. 22, 682 (1989)] is the electro-optic coefficient, and $t_{\text{GaP}} = 0.4769$ is the Fresnel coefficient for reflective loss at the GaP crystal. We have included this information in the revised Methods section. We have also added the two plots below (EO sampling traces and individual scans with opposite helicities) in the revised version of the Extended Data Figure 1 and Figure 3

Electro-optical sampling traces.

Individual helicity scans, their difference and Fourier transform of the difference.

2. The description on the comparison of two peaks in Fig. 2b is quite terse. Compared to the dynamical multiferroicity model predicted by Juraschek et al., the amplitude of the high frequency peak is significant as observed experimentally. While the authors associate this to the electronic nonlinearities, they do not explicitly mention the microscopy of this electronic nonlinearity. They mention its associated with third-order susceptibility, but this does not clarify at all.

We agree with this comment and we apologize for the bad timing of the second arXiv manuscript where all these details are found. There are two contributions to the total response besides the dynamical multiferroicity: (i) the terahertz electronic Kerr effect (EKE) [Appl. Phys. Lett. 95, 231105 (2009) and Nature Photonics, 4, 131-132 (2010)], i.e. the THz counterpart of the optical Kerr effect, due to off-resonant electronic transitions and (ii) the ionic Kerr effect (IKE) [arXiv:2210.14053], which exploits the ability of strong THz light pulses to drive a nonlinear excitation of multiple infrared-active phonons. For further details, please refer to our study on the experimental and theoretical description of the full nonlinear optical response of SrTiO₃ which is now available on [arXiv:2210.14053]. The crucial point relevant to our work is that neither the EKE nor the IKE can explain the relative amplitude of the measured low frequency peak with respect to the high frequency peak.

3. An apparent question that comes to my mind: What happens to the difference Kerr signal as the mode systematically softens to 0.2 THz at 5K. What happens to the ratio of the a" and a...?

We thank the referee for this question. At a temperature $T = 110\text{K}$, the material undergoes a cubic-to-tetragonal phase transition. Hence, at low temperatures we expect a change in the components of the nonlinear susceptibility tensor mediating both the electronic and the ionic Kerr effect and a consequent change in the shape of the sample response. Moreover, in the tetragonal phase, SrTiO₃ presents additional Raman active phonon modes which can couple to the excited infrared-active soft phonon, see [Li, X. et al. Terahertz field-induced ferroelectricity in quantum paraelectric SrTiO₃, Science 364, 1079–1082 (2019)]. This further complicates the analysis, and we decided not add complexity to the one already present. A detailed discussion is out of the scope of the present work, which focuses on the cubic phase of STO. We plan to address this in a future study.

4. It is not clear to me why the authors state that the quadratic dependence of a... mode on the driving THz field is an indication of extra contribution and how is it related to electronic nonlinearity?

We thank the referee for raising the issue. In general, for a time-domain signal $f(t)$ whose amplitude is proportional to E^2 , the amplitude of its spectrum is proportional to E^2 as well:

$$f(t) = x(t)AE^2 \Rightarrow |F[f(t)](\omega)| = |F[x(t)](\omega)|AE^2,$$

where F is the Fourier transform and $A > 0$ is a generic proportionality factor which keeps the measurement units consistent between the left and right hand sides of the equations. In general, the quadratic scaling of the Fourier transform as a function of E depends on the spectral weight at a given frequency. Hence, for two different frequencies ω_- and ω_+ , one has two different curvatures $c1 = |F[x(t)](\omega_-)|A$ and $c2 = |F[x(t)](\omega_+)|A$. However, if one normalizes the Fourier transform in the conventional way, i.e. frequency component by frequency component, one expects the same curvature, determined by the common factor A that scales the entire signal $x(t)$. This is indeed the type of

behavior expected when a single mechanism with this type of scaling is at play. In the now available Ref. arXiv:2210.14053, we identify this mechanism in the Kerr effect, electronic and ionic, the latter explained in point 2 above, a nonlinear optical effect analogous in symmetry to the well-known electronic Kerr effect, but in this case mediated by the ionic motion induced by the terahertz light. The ionic Kerr effect predicts a common scaling factor A for both the low and the high frequency components. However, experimentally, we observe that the curvature of the E -field dependence has an additional contribution B for the low-frequency mode only, for a total curvature $A + B$. This additional contribution to the curvature B must come from a different mechanism which does not have high frequency components, such as dynamical multiferroicity.

5. Lines 205-206, has very little grounding, being a speculation. I am not sure if I completely understand what the authors are trying to imply here. How does the transfer of angular momentum to valence electrons leads to a larger enhancement and why only the high frequency mode? I fail to find to good train of thoughts that connects it all. A careful rephrasing and proper justification (instead of speculation) would be crucial for the said claim.

We agree with the referee that the statement was too speculative. Given the limited amount of words available, we removed that statement, and will address the issue in detail in a future work.

6. How does the overall result apply to materials other than STO? The conclusion is too much focused around STO. In addition to the proposed possible implications of their results, the authors need to bring out the generality of the obtained results.

The results shown here are expected to apply to all systems with infrared-active phonon modes at the center of the Brillouin zone. Ref. [Phys. Rev. Materials 3, 064405 (2019)] lists 35 materials, such as Rocksalt (CsH, MgO), Wurtzite (InN, CuH) and perovskites compounds (KTaO₃, LiTaO₃, BaTiO₃, CsPbF₃). KTaO₃ was theoretically studied in [Phys. Rev. Res 3, L022011 (2021)], and we have seen preliminary results in conferences, showing similar results to ours, clearly with the resonance at a different phonon frequency. We now added one sentence about this in the revised version of the manuscript.

7. Minor one: In the figure caption of Extended Data Fig. 2, the authors mentioned about oscillator strength found in Ref. 44. I think it is typo, I did not find any such expressions in Ref. 44, did they mean Ref. 43 instead?

We thank the referee for noting the typo. We corrected the reference number, whose overall numbering has changed in the revised version of the manuscript.

Referee #3 (Remarks to the Author):

Summary of key results:

Basini et al. report the observation of an optical polarization rotation in SrTiO₃ driven by circularly polarized THz light pulses that exhibits some interesting features. Most notably, the pump-helicity dependent component of the polarization rotation has frequency content at roughly zero frequency and twice the driving frequency (indicating a response quadratic in the driving field), and the magnitude of the response shows a non-monotonic temperature dependence (peaking close to the temperature where the THz pump is in resonance with the soft polar phonon of SrTiO₃). Based on comparisons to the theory of “dynamical multiferroicity,” the authors attribute part of this signal to a magnetic moment arising due to circulating ionic motions from the resonant phonon excitation (with the other part being explained by a “trivial” nonlinear optical response). Remarkably, the induced moment deduced from these measurements is found to be four orders of magnitude larger than theoretical predictions.

Originality and significance:

This work contributes to the growing field of ultrafast optical control of materials in general, and of “nonlinear phononics” (in which resonant vibrational control is leveraged to create non-equilibrium structures and phases), in particular. The idea of “dynamical multiferroicity” upon which this experiment is based was proposed in 2017. It essentially rests on the phenomenological notion that the time-varying polarization associated with charged ions moving in closed loops acts like a magnetic field and can generate an internal magnetization, even in an otherwise non-magnetic material. The effect arises naturally from Maxwell’s equations (it’s basically a statement of Ampere’s Law). It can also be considered the ionic equivalent of the inverse Faraday effect, which is more commonly realized at optical frequencies (as pointed out by the authors in the main text), but the difference in frequency does not preclude the two phenomena from having the same origin. From that standpoint, I don’t even see the underlying phenomenon that the experiment seeks to demonstrate particularly novel or as providing a meaningful advance in our understanding.

On the other hand, from a practical point of view, it is interesting to explore new ways to control materials’ properties on ultrafast timescales using light. However, the principle of resonantly driving circular atomic motions to create effective magnetic fields was already experimentally demonstrated by Nova et al. in 2016 (Ref. 21). The new aspect of the work by Basini et al. is to apply this technique at low frequencies in a non-magnetic material. These aspects don’t provide any new insights in my view. It is somewhat intriguing that they observe such a large induced magnetic moment in their experiment, but I have my doubts about the technical aspects leading to their conclusions (see below). Even putting those doubts aside, though, based on the considerations above, both in terms of the theoretical framework and experimental validation, I do not consider the results presented in this paper to be highly original or significant.

We thank the referee for his or her work in revising our work. Even in this case, we refer to our introductory comment on the interpretation of a generic “inverse Faraday effect”. We clearly disagree with the overall opinion of originality and significance. However, given that they build up on the point-by-point issues below, we prefer to first address each of them in detail, and then rebate to the more general statements.

Data, methodology, conclusions:

The most important experimental conclusions the authors extract from their data are that (a) there is a magnetic contribution to their signal which arises from the dynamical multiferroic effect, and that (b) this magnetic contribution is much larger than expected indicating some additional mechanism at play.

Making such statements from the optical data is highly non-trivial, and upon thoroughly examining the experimental data and methodologies presented in the main text and supplement, I do not think there is sufficient experimental evidence to support their claims. Below I detail some of my technical concerns:

- Is the THz excitation really circular? Is the helicity dependence accurately measured?

The authors use a bandpass filter and a quartz birefringent waveplate to create circularly polarized pulses centered at 3 THz. How confident are the authors that the output of the waveplate really produces circular light? Even at IR and visible wavelengths where optics are typically of much higher quality and better characterized, slight misalignments or non-ideal optical elements can lead to differences in the polarization state. Indeed, in the supplement, the authors show EOS traces of the E_x and E_y components which have obviously different FFT spectra, which indicates that the THz light cannot be circular (would rather be somewhat elliptical). The authors also do not mention how they measure the E_x vs. E_y components nor do they mention how they obtain the opposite THz helicities. Presumably the opposite helicity is achieved by rotating their quarter waveplate by 90 degrees and remeasuring (please correct if I'm wrong, more details would be appreciated). Again, even if the waveplate was ideal, it is unlikely that this rotation can be done with perfect precision. These non-idealities will lead to a number of artifacts, one of which is some non-canceling linear components of the signal such that the polarization rotation they measure does not reflect a "MOKE" signal.

We thank the referee for this comment. A detailed description of our protocol for characterizing the polarization of the THz excitation should suffice to answer the questions. We omitted it in the original manuscript due to space limitations, but we include it here and partly in the revised supplementary material.

The device for producing circularly polarized pulses is a quartz birefringent quarter-wave plate (B-QWP) from TYDEX, whose thickness is tuned to provide a $\pi/2$ phase shift at 3 THz. For a broadband pulse, the B-QWP introduces a linear phase as a function of frequency, which translates (in the time domain) into two shifted replicas of the input linearly polarized pulse along the crystalline axes of the B-QWP. The different FFT spectra for the E_x and E_y components are then a consequence of projecting the replicas along two perpendicular axes rotated by 45 deg with respect to the B-QWP crystalline axes. Indeed, as correctly supposed by the referee, opposite helicities for the THz excitation have been obtained by rotating the B-QWP at ± 45 deg with respect to the systems of reference of our experiment (which includes the xy -plane). The relative rotation of 90 deg was controlled with a motorized stage, with a precision of 0.5 deg. For broadband light, this procedure does not produce perfect circularly polarized light. However, in all the experiments presented we had a bandpass filter between the broadband field and the sample to restrict the bandwidth close to the target 3 THz frequency, for which the B-QWP operates according to its design. Nonetheless, we stress the fact that we don't need perfect circularly polarized light, since the induced magnetization in the dynamical multiferroicity picture is proportional to the $\vec{P} \times \partial_t \vec{P}$ term, which is different from zero also for an elliptically polarized driving force.

In any case, we agree that the characterization of the THz light emerging from the B-QWP is necessary in order to validate all the involved technical details, and to record the actual pump traces for the modeling. We have added a description of the characterization protocol in the revised Methods section, and the relevant figures in the Extended Data.

The electric field components E_x and E_y have been measured by electro-optical sampling in a 50 μm -thick (110)-cut GaP crystal, and the time-domain traces for opposite helicities (labeled as LCP and RCP for simplicity) are shown both in the reply to comment (1) raised by Referee #2. Both field components can be acquired, in subsequent measurements, by rotating the GaP crystal by 90 deg around the light propagation axis. This operation is enough to get sensitivity to the two orthogonal components of the THz pump, as shown in Ref. [46]. To get quantitative feedback on the involved polarization states, we calculate the Stokes parameters since, from the time-domain traces, we are able to access amplitude and phase as a function of frequency for the E_x and E_y pump electric fields. For each frequency component ν_i , the $S_{0,i}$, $S_{1,i}$, $S_{2,i}$ and $S_{3,i}$ parameters allow us to unambiguously identify the polarization state. The figure below shows the Stokes parameters we obtain for our experiment.

A change of sign for the S_3 parameter is an indication of opposite helicities. In case of a broadband pulse, the following equation holds (H. C. van de Hulst, “Light scattering by small particles”, Dover, 1981):

$$(Q/I)^2 + (U/I)^2 + (V/I)^2 \leq 1, \text{ where } I = \sum_i S_{0,i}, Q = \sum_i S_{1,i}, U = \sum_i S_{2,i} \text{ and } V = \sum_i S_{3,i}. \text{ The } (V/I)^2$$

quantity can be considered as an indication of the average “amount” of circular polarization we have in our THz pump pulse. Considering only the 0.5 THz FWHM region of the peak, we find that the beam is 85% - 90% circularly polarized, see the Table below.

	$(Q/I)^2$	$(U/I)^2$	$(V/I)^2$
Ideal case	0	0	1
RCP	0.009	0.020	0.859
LCP	0.001	0.004	0.892

• How is the field strength calibrated?

The authors state in the supplement “The exact size of the beam is not crucial to estimate the fluence since we characterised the electric field of the radiation.” If the EOS is properly calibrated, this can certainly be true. But, then when discussing the EOS in the methods “In panel (a) the reflected terahertz field from the back side of the GaP crystal starts to interfere with the direct beam at approximately $t = 3.8$ ps. This complicates the electro-optic sampling data presented here...” So, with these apparent artifacts in the EOS data it is unclear to me how the authors actually calibrated the field strength in their experiment. Also, the correspondence between the field measured by EOS and the field incident on the sample requires that the EOS crystal be placed in the exact same position as the sample. Was this done and to what accuracy?

There are no error bars anywhere in this paper, which to me is a huge huge red flag.

We thank the referee for this comment. We refer to the answer to Comment (1) by Referee #2 who asked the same question, and we present the EO sampling characterization there. We respectfully disagree with the comment about the lack of error bars being a red flag. We admit that we overlooked this matter, and we amend this here, but the reason they were omitted is simply because the error bars were mostly very small and just contributed to clutter of the figure. In the long process of writing the manuscript, at some point we forgot to put them back in. We thank the referee for pointing this out, which allows us to fix this in the revised version of the manuscript. We now included the standard error in all experimental figures. Each trace presented in the manuscript is an average of multiple scans (from 2 to 20, depending on the signal level), but the main features are already visible after a single scan with standard lock-in settings (300 ms integration time per point, 500 Hz lock-in frequency). The noise in our setup in this configuration is below the microvolt level for the single acquisition, and the typical peak signal exceeds 10 microvolts. It is a rather large signal compared to other signals that are routinely measured in pump-probe experiments in our lab, which can be of the order of hundreds or even tens of nanovolts. For the EO sampling traces, the lock-in signal is stable to three or more significant digits at a given time step since that signal is in the millivolt range, and hence much lower gains are needed. In those data, the error bar is smaller than the line itself. We agree with the referee though that we should have been more pedagogical in explaining this.

• What about other contributions to the signal? (Related to previous questions)

One of the most important the authors make in trying to interpret their data as an induced magnetic moment is that when comparing the measured pump-induced polarization rotation to the product of the E_x and E_y fields, there is a difference in the low-frequency peak of the spectrum. That is, if one assumes that there is some “trivial” third-order nonlinear optical effect that can also create a polarization rotation (i.e. pump-induced birefringence), then the authors say that this effect should be proportional to $E_x \cdot E_y$, and any part of the signal which is not strictly proportional to this product is magnetic. I have many problems with this line of reasoning.

First of all, the authors do not even provide a cursory argument in the paper for why the third-order optical effect should go like $E_x E_y$ nor do they account for the strong polarization dependence expected from this response, simply referencing a separate study. The problem is that this separate study doesn't exist yet! To me, it is totally unacceptable to defer a key aspect of the data analysis to a paper which is still in preparation. For argument's sake, let's assume the authors' assumptions are valid, then the calibration and measurement of the exact fields E_x and E_y becomes extremely crucial, but based on the data and method details presented, the accuracy of these fields are highly questionable.

We agree with the referee, and we apologize again for the bad timing of putting the related work out a couple of weeks later than planned. It was obviously not the intention, but we had discovered an important last-minute issue that we wanted to clarify before putting the manuscript online. We urged as much as we could, knowing that we had a paper out referring to it, but obviously it took too long. The manuscript was available about a month later on the arXiv [arXiv:2210.14053] and is currently under review. In that work, we performed an extensive study on the experimental and theoretical description of the full nonlinear optical response of SrTiO₃ in the cubic phase. We show there that the nonlinear optical response of SrTiO₃ is due to a Kerr effect which includes two contributions. In addition to the well-known electronic Kerr effect (EKE) caused by off-resonant electronic excitations in wide-band insulating STO [Appl. Phys. Lett. 95, 231105 (2009)], some of us have shown that a ionic contribution, associated with the nonlinear excitation of two TO₁ phonons, is present. As detailed in Ref. [arXiv:2210.14053], such an ionic Kerr effect (IKE) should be taken into account to fully reproduce the temporal shape of the signal, in particular the up-down behavior noted by the Referee in one of the comments below. The response proportional to $E_x E_y$ is instead due to the well-known EKE for circulatory polarized pump fields. Intuitively, this is equivalent to the E^2 dependence for a linearly polarized field.

We now added an even more detailed characterization of the experimental THz fields in the revised Extended Data. However, we respectfully disagree with the referee that the accuracy of the fields in the original manuscript were "highly questionable". This is simply a broad statement not backed up by a factual argument. The field E_x and E_y were carefully characterized and shown already in the original manuscript. The characterization of the electric field by means of electro-optical sampling (stated clearly in the original manuscript) is the same used by the terahertz community for more than two decades. Sometimes figures or explanations are omitted because they are simply deemed to be obvious by the research community. One could argue on whether other choices could be made, but we would be grateful for a more neutral approach. Many of us performed state-of-the-art and pioneering terahertz experiments for at least a decade, documented by our publication record in the field. Characterizing a terahertz field, while still more complex than simply measuring the power of a visible laser, it is a routine measurement in many labs around the world. We think that in a high-impact journal, room should be left for original measurements, not routine ones.

Secondly, the authors do not try to rule out any other possible contributions to the low-frequency peak before attributing it to magnetism. For example, in the text they point out "intense coherent THz excitation of the soft mode in STO can lead to highly nonlinear phonon responses that overcome the quantum fluctuations creating a ferroelectric order absent at equilibrium" (Ref. 38). There was also other work that showed long-lived ferroelectric states upon phonon pumping (Nova et al. Science, 2019 <https://doi.org/10.1126/science.aaw4911>) which should be cited. The experimental work here uses intense coherent THz excitation, so we might expect ferroelectric order to develop in the same way as these

works. Obviously, such a phase transition could drastically change the optical properties in a highly nonlinear way, including the reflectivity/refractive index at the probe frequency, which could lead to spurious low frequency content in the polarization rotation (I will also come back to this point later). There are other possible artifacts that must be ruled out as well, including polaritonic effects, higher order nonlinearities, etc.

We thank the referee for the comment, which helps to clarify the context of our work. Given the similar order of magnitude of the electric field amplitude used with respect to Ref. [Li, X. *et al.* Terahertz field-induced ferroelectricity in quantum paraelectric SrTiO₃, *Science* **364**, 1079–1082 314 (2019)], it is worth considering the possibility to be close to the ferroelectric transition observed in the same reference. Anyway, as already pointed out in the same work, no symmetry breaking due to the emergent ferroelectric order is observed above 60 K. Moreover, in a previous study performed on SrTiO₃ at 100K, the lattice response to a similar THz driving field was directly mapped by means of femtosecond X-ray pulses [Nature Physics, 15, 387-392 (2019)]. A direct visualization of the atomic displacement evidenced nonlinearities in the oscillator response, but no signs of a phase transition. Because of this, in the high temperature range of our measurements (160 K-375 K), we ruled out the possibility that SrTiO₃ undergoes a ferroelectric phase transition. Regarding the other three aspects pointed out by the referee: (i) the additional reference, (ii) polaritonic effects and (iii) higher order nonlinearities:

(i) We agree on the relevance of the suggested reference, and we now added it in the main text.

(ii) Polaritonic effects associated with the excitation of an IR-active phonon can be observed in THz pump-optical probe measurements in non-centrosymmetric systems, where the IR-modes are also Raman active, thus allowing for a linear coupling with the THz pump and a nonlinear (two-photon) coupling with the optical probe pulse. The simultaneous excitation of the upper and lower phonon-polariton branches would result in the observation of a beating in the time domain signal, as reported e.g. in Ref. [Sci. Rep. 6, 38264 (2016)]. Nonetheless, these effects are not expected in centrosymmetric SrTiO₃, since the soft-phonon mode is IR but not Raman active.

(iii) Given the inversion symmetry of the material, we do not expect any contribution from a fourth order nonlinear response (i.e. $\chi^{(4)}=0$). We rule out a detectable contribution due to even higher order nonlinearities, given that they are expected to be several orders of magnitude lower than the third order nonlinearity [R.W. Boyd, Nonlinear Optics, Chapter 1, 3rd edition, Elsevier Inc. (2008)].

One measurement which would be very valuable in this context is probe polarization dependence, which should exhibit very different behavior if the signal arose from magnetism vs. alternative optical effects.

Has this been done?

We thank the referee for this important observation. We had already measured, and now remeasured after this request (in order to have a comparison close in time), the sample response by means of P and S probe polarizations (at 400 nm), see the figure below. Panel (a) shows the traces recorded with a P polarized probe for the two helicities of the THz pump field. As for all the measurements reported in the manuscript, the balanced detection includes a half-wave plate, to gain sensitivity to polarization rotation. Panel (b) shows the same traces, but recorded with a S polarized probe. After switching the probe polarization from P to S, we kept the same orientation for the half-wave plate, in order to ensure consistency between measurements. Indeed, the half-wave plate orientations for the initial balancing of the detected intensity are the same for both probe polarizations.

If we compare the data with the same helicity (sharing the same color in the plot), we can observe an overall change of sign between them. The different absolute reflectivity of the probe signal between P and S polarizations at 45 deg of incidence makes a quantitative comparison challenging, but such a comparison is not necessary. The experimental evidence clearly indicates that we cannot decouple the dynamical multiferroicity signal from the electronic nonlinear response associated with $\chi^{(3)}$ just by comparing the measurement with perpendicular probe polarizations.

While the change of sign for the $\chi^{(3)}$ response is easy to visualize, we also independently verified the behavior of the magnetic signal by measuring the static Faraday effect in STO. The figure below shows the amount of Faraday rotation experienced by the transmitted 400 nm probe beam when a magnetic field is applied in the direction of the probe propagation and perpendicularly to the sample surface. The linear dependence between the applied field magnitude and the rotation angle allows us to estimate a Verdet constant of approx. 180 rad/(m·T). The change of sign when the polarization is switched from S to P is consistent with a signal of magnetic origin. We added this characterization in the Extended Data.

• How is the magnetic moment obtained?

Again, one of the key claims is that the magnetic moment observed is four orders of magnitude larger than what would be expected from theory. There are several aspects of this conversion that need clarification.

First, are the authors considering only the “extra” magnetic part of the low frequency signal when extracting the magnetic moment? If so, how can this reliably be done without having the proportionality factor relating $E_x \cdot E_y$ to the nonlinear optical background that is supposed to be removed?

Indeed, the referee is correctly guessing that we consider only the “extra” magnetic part. To extract the magnetic moment, we normalized the nonlinear optical response on the measured signal, so that the peaks at $\omega_+ = 6$ THz overlap (as explained in row 92-96 and 127-129 of the original manuscript). This is justified by the evidence that (i) the shape and amplitude of the peak at ω_+ can be reproduced by the calculations based on the symmetry of the $\chi^{(3)}$ tensor, see Ref. [arXiv:2210.14053] and (ii) no contribution from the dynamical multiferroicity is expected at 6THz [Phys. Rev. Materials 1, 014401 (2017)]. We kept an identical approach when evaluating theoretical and experimental responses, given that the units of the two measurements are different and scaling factors are arbitrary. We then ascribed the different amplitude in the low frequency peak ω_- to a dynamical multiferroicity contribution and converted the correspondent rotation amplitude into a magnetic moment (see Methods).

Second, the authors take the literature value for the Verdet constant from transmission measurements and use it as their MOKE calibration. This is not a sound way to obtain the magnetic moment from MOKE data. The authors note that they measured also in transmission and only saw differences up to a factor of two, but this does not mean that the literature Faraday coefficient can be used. The Verdet constant (both magnitude and sign) will be highly sensitive to wavelength near the band edge where the probe frequency is parked in this experiment, for example. So, one really needs in the best case scenario to directly measure the static MOKE signal as a function of field to have an accurate calibration on the actual sample using the real experimental optical parameters. Barring that, one at least needs to also take into account changes in reflectivity and in the magneto-optical coefficients that might also be driven by the THz field. As mentioned above, these effects can be significant since it's known that STO is highly polarizable by the THz field and can even undergo a phase transition, which would drastically change the optical spectra. The referee is correct that the amplitude of the Verdet constant V is highly sensitive near the band edge; however, we note that we are not too close to the band edge to expect a sign change or a remarkably different response. A change of sign and a lower value of V is on the other hand expected at around 800 nm, see the table below from Ref. [Weber, M. J. Handbook of Optical Materials (sec. 1.6), CRC Press LLC, Boca Raton (2003)]. The choice of probing at 400 nm was motivated by this very consideration. In addition (see response to the comment above) we have now determined the Verdet constant in our own samples, and found a value of 180 rad/(m T) at 400 nm, within 20% of what was measured almost 50 years ago on a different sample. This seems to confirm the robustness of the literature values for this material for static measurements.

{REDACTED}

As for the THz-driven data, as already pointed out Ref. [Li, X. *et al.* Terahertz field-induced ferroelectricity in quantum paraelectric SrTiO₃, *Science* **364**, 1079–1082 314 (2019)], no symmetry breaking due to the emergent ferroelectric order is observed in STO above 60K. Moreover, in Ref. [Nature Physics, 15, 387-392 (2019)], the STO lattice response at 100 K to a similar THz driving field was directly mapped by means of femtosecond X-ray pulses. A direct visualization of the atomic displacement evidenced nonlinearities in the oscillator response, but no signs of a phase transition. Hence, in the work presented here, where we considered a STO sample in the cubic phase, at 300K and with lower THz fields, we can safely rule out the possibility that the material undergoes a ferroelectric phase transition. One can also refer to our measurements as a function of temperature: if a change in the optical spectra would occur while lowering temperature, a remarkable change in the sample response should be observed. This is however not the case, see Figure 3a of the manuscript as well as the one that we plot here below. This latter figure shows the time domain response (plotted as the difference between the signals obtained with RCP and LCP pump fields) that we measured at 160 K and at 375 K, i.e. the minimum and maximum temperature of our measurements. The response is qualitatively the same.

Thirdly, the extracted moment is highly dependent on the estimated pump-probe penetration depth mismatch, which may also exhibit nonlinear and time-dependent behavior upon pumping.

As in the reply above, we note that with a probing wavelength of 400 nm (3 eV), we are still in the transparency region of STO (3.7 eV direct bandgap), i.e. with a probe penetration depth exceeding the millimeter range, while the penetration depth of the terahertz pump field is on the order of 1 micrometer. Hence, we are in the limit of a total mismatch between pump and probe, which does not lead either to convolution artifacts when estimating quantities. We are substantially integrating with the probe over the whole thin surface region of the STO excited by the THz field, with a constant probe intensity throughout the sample.

- Other issues

If I assume that the effect is driven by a phonon-induced magnetization, I would expect from the equation $M \sim P \times dP/dt$ that the signal should have a rectified contribution and a contribution at twice the phonon frequency. But, in the temperature dependence of the signal, a peak is always found in the spectrum at twice the THz frequency (6 THz) even though the phonon frequency is changing a significant amount over that range. Why are features not seen at twice the phonon frequency instead?

The dynamical multiferroicity signal $M \sim P \times dP/dt$ has indeed a component at twice the phonon frequency, if the two phonon modes are degenerate, as in our case. However, the amplitude of this component is also expected to be negligible for degenerate phonon modes, see Eq. (3) and (22) of Ref. [Phys. Rev. Materials 1, 014401 (2017)] and the discussion in row 116-118 of our original manuscript. We stress again that the observed strong 6 THz component that we see in all traces is *not* due to dynamical multiferroicity, but to the nonlinear optical response of SrTiO₃ to the 3 THz pump field. Hence, its frequency is expected to *not* depend on temperature.

The calculated moment time dependence Fig. 4a only shows a single dip, while the actual signal clearly shows a bipolar behavior where it has a positive rectified component around 0.5 ps and the negative part around 1-1.5 ps. What is the origin of this “up-down” behavior and why is not reproduced by the theoretical calculation?

The referee is correct in identifying the up-down behavior. We ascribe the “up” part to the ionic Kerr effect, as described in Ref. [arXiv:2210.14053], and the “down” response to dynamical multiferroicity, as discussed in this work. We can reproduce both behaviors if we include the full calculations considering the EKE and IKE responses, as done in the now available reference and as we show in the figure below. In the original manuscript, we decided not to add this further discussion to a manuscript already dense in content, but we have now included a short sentence about this aspect.

Even if I take the arguments in the paper at face value, and I assume that the ionic motions really are creating an effective B-field, I have trouble understanding where the resulting magnetization comes from. SrTiO₃ is a d⁰ compound. There are no electrons in the d-shell which can be polarized by the effective field. If the idea is that angular momentum is being transferred from the phonons to the electronic system, can the authors explain what electronic system is being coupled to?

We agree with the referee that there should not be a coupling channel with the electronic system while in equilibrium. However, this is not an equilibrium experiment, with clear signatures of nonlinear dynamics, as discussed in depth even in this rebuttal. The effect emerges due to the orbital motion of ions. At the moment, we can only speculate on possible mechanisms, and we have identified two. The first, is that the observed magnetism is due to the orbital degree of freedom of electrons, something that is only recently being discussed in literature. Magnetization in dynamical multiferroicity mechanism appears as a result of dipole moment dynamics. The titanium atom is charged, it is in a 4+ state, (Sr is in 2+ and O in 2-). Hence, the motion of the Ti ion induced by excited phonon mode will result in orbital magnetism. But the referee is implicitly asking a valid question: how exactly the transfer of angular momentum in the two cases would occur. Detailed analysis of this phenomenon is beyond the scope of this work. A second possible mechanism is the modification of the band structure due to the phonon dynamics, which could be large enough to possibly induce a spin splitting of the bands, recently proposed [R. M. Geilhufe and W. Hergert, Phys. Rev. B 107, L020406 (2023)]. As clearly stated in the original manuscript, only a free electron laser measurement with circularly polarized photons will be able to reveal the path of angular momentum transfer. We prefer not to dig into these speculations for two reasons. First, they are speculations and would occupy a space of the text that we need to use to explain other and more useful details of our results. Second, we performed state-of-the-art measurements with clear experimental signatures of a signal consistent with dynamical multiferroicity, and state-of-the-art theoretical calculations with finite temperature effects which reproduce many features of the experiment. It is beyond the scope of a single work (which also discovered a new effect, the ionic Kerr effect, topic of yet another paper) to dig further into this aspect. Furthermore, while identifying several potential issues, which we have all addressed, none of the referees suggested an alternative explanation to our data. Finally, the recent results suggested in our original manuscript demonstrating magnetization switching due to

circularly polarized phonons, is a strong proof of the validity of the proposed Barnett effect, and that indeed a large angular momentum is transferred to the electronic system.

The raw data, in addition to the helicity dependent data should be given somewhere, so readers can assess the data analysis better. It would also be helpful to have the pump e-field superimposed on top of the rotation signal.

We completely agree with the referee. We had long discussions among ourselves about what to include and what not due to the already extended length of the manuscript and related material. In the revised Extended Data Figure 4, we have included the raw data with superimposed the two components of the pump E field, as suggested. Here below is that data for the Referee to see directly.

The fact that the signal gets largest when the phonon is in resonance with the 3 THz pulse is not necessarily evidence that the signal comes from this dynamical multiferroicity – even a “trivial” nonlinear optical effect will see a resonant enhancement of this type.

We respectfully disagree with this comment, and we do not really understand it, given that already in the original manuscript we presented the resonant effect which is *not* due to dynamical multiferroicity. In many complex physical systems, the “trivial” assumption is that one can study the electronic, magnetic or phononic independently, given that the degrees of freedom are well separated. If there are two systems

involved (like the electronic and the phononic one), a coupling between them needs to be demonstrated, or there is no a priori reason to expect that a phonon resonance will propagate into the electronic system. Our measurements only allow us to look at the electronic response. Hence, the coupling with an underlying infrared-active phonon mode is not necessarily true. In the now-available complementary manuscript Ref. [arXiv:2210.14053], all this is discussed in greater detail, and that work further corroborates this picture. However, already in the original manuscript related to this rebuttal, we discussed in depth that there are two distinct effects at play. First, the nonlinear optical response, with two distinct frequency peaks of similar amplitude, quadratic field dependence, and resonant temperature dependence. Second, the dynamical multiferroicity effect, identified as the “extra” weight at the low frequency peak, which does not have a 6 THz component, and with a characteristic field and temperature dependence.

Final comments:

To reiterate: this paper presents an attractive picture of using circularly polarized light to create magnetization in a non-magnetic material by driving “rotating” phonons. On the technical side, the evidence for this picture is highly circumstantial, and I am not convinced that the analysis is sufficiently comprehensive and that all relevant artifacts have been ruled out. More generally, in the context of the field of optical control/nonlinear phononics, I do not think the results of this paper constitute a highly original or significant contribution. While I do think the work presented here is valuable in a more narrow context and should eventually be published somewhere, in my opinion, it is not appropriate for Nature, and it would certainly need more comprehensive data analysis and some additional experiments as presented above before publication anywhere.

We believe that we have answered in depth to all of the issues raised by the referee. We strongly disagree with the fact that the picture is “highly circumstantial” and that there are unspecified “artifacts”. Those are broad statements, made without providing any alternative interpretation. We hope that our thorough work in replying to all of the questions raised by the referee will change his or her opinion on this matter. We are grateful to him or her for having found unclear aspects, which gave us a chance to amend parts which could have been presented or explained better. However, we stress that no new suggested measurements or analysis that we performed as requested has in any way modified the original statements and interpretations, and no experimental or theoretical errors were found. Furthermore, the very recent results that appeared in Ref. [arXiv:2305.11551] verified one of the applications that we had anticipated in the original manuscript, namely the switching of a thin film ferromagnet on top of a substrate where circular phonons can be induced. This is an additional, strong experimental proof, that a magnetic moment of the order of the Bohr magneton can be induced by phonons in the way we proposed it, likely via the Barnett effect. Finally, we completely disagree that inducing a large magnetization in a non-magnetic dielectric, using light resonant with the phonon dynamics, is not a highly original nor significant contribution. It is much more significant than enhancing magnetic order in an already magnetic material, as the number of potential materials to be exploited is much larger. The referee has not provided references to previous works where intense resonant terahertz fields, circularly polarized, were used to trigger phonon dynamics. There is now another one on the arXiv, months after our original submission, proving an important application of the effect based on the mechanisms that we originally proposed. Hence, it is not clear to us on what basis originality is under discussion. As for the significance, the two other referees found our work of extreme significance. Furthermore, the ultrafast community has granted us several invited talks to present these results at the major international conferences, including the APS March Meeting 2022 and

the Gordon Research Conference on Ultrafast Phenomena in Cooperative Systems 2022, and recently received an invitation for the same GRC in 2024. We respect the opinion of the referee, but it is clearly not aligned with the one of many in the physics community at large.

Reviewer Reports on the First Revision:

Referees' comments:

Referee #1 (Remarks to the Author):

The authors have revised their manuscript and, in my opinion, clarified the points raised by me and the other two referees appropriately. In particular, the clarification of the relation of the proposed mechanism of dynamical multiferroicity in relation to the inverse Faraday effect is very much appreciated, as well as the explicit inclusion of the raw data of the Kerr measurements.

Concerning the validity of the experiments (point raised by referee 3): I have seen additional experiments recently, specifically [arXiv:2306.03852] and a conference presentation from a completely different research group two weeks ago, which corroborate the measurements in the present manuscript. Together with the clarifications provided by the authors in their rebuttal, I have no doubt left that the results that the authors present here are indeed a phonon-based mechanism.

Concerning impact (also point raised by referee 3): Research on chiral phonons has experienced an unprecedented boom within the past 12 months across different subfields within condensed matter physics and materials science, and the results presented by the authors mark the first experimental realization of chiral phonon-induced magnetism. In my opinion, this study already now has great impact on the field, as evidenced by the increasing amount of follow-up work by other research groups that has surfaced during the past months.

I therefore recommend publication of the manuscript with no further changes.

Referee #2 (Remarks to the Author):

The authors have satisfactorily answered all my concerns. They have presented a detailed analysis procedure that I requested, which now validates the results and the conclusions drawn from these experiments.

As mentioned in my earlier review, this work is an important addition to the field of THz control of novel phenomenon and deserves a publication in Nature. I can now recommend its publication.

Referee #3 (Remarks to the Author):

The authors have presented a revised manuscript with an extensive amount of extended data and supplementary information added. They have included more details about the pulse characterization methods. They have also added raw data, error bars, and new measurements of the probe polarization dependence, which are very helpful in understanding the signals they observe. I appreciate the authors thorough work, and I do think the revised manuscript is much improved. However, I still have a number of concerns regarding the analysis and interpretation of the data that would prevent me from recommended the paper for publication. My main concerns are described below in no particular order.

1. As I stated in my initial report, the pump-probe penetration depth mismatch is a crucial factor

in extracting the overall size of the magnetic moment that is extracted from these measurements. The authors argue that the probe wavelength is in the transparency region for STO, so that it penetrates the entire sample, while the pump penetration depth is on the order of a few microns at all measured temperatures. I'm not sure how the authors come to this conclusion regarding the probe penetration. Looking at measurements of the absorption coefficient near the band edge of STO [Gogoi et al., PRB 93, 075204 (2016)], the band gap at 300 K was measured to be 3.15 eV – quite close to the probe photon energy at 400 nm (3.1 eV). In addition, a strong absorption tail at lower energies was found. Extracting the absorption coefficient from these measurements seems to give a probe penetration depth of ~ 3 microns or so, roughly the same as the pump penetration. This contradicts the authors' assumption that the probe penetration is on the order of millimeters and will scale the extracted magnetic moment accordingly.

2. Related to the above observation that probe photon energy is, in fact, near the band edge, the value of the magneto-optical coefficient can be modified by the pump field. In transition metal oxides, magneto-optical transitions are often associated with vibronic transitions (especially at energies close to the band edge), which are strongly coupled with lattice excitations. Since the authors are directly driving a phonon with their pump, it is, to me, not unlikely at all that the magneto-optical coefficient may be altered as a result of this driving. I want to stress that this is not just a pedantic point – such effects are present and affect magneto-optical measurements even in well-established magnetic compounds. In this case, where the origin of magnetism is less direct, I think it is important that the authors are able to address this point and rule out such a scenario.

3. The authors show that the peak at 6 THz and the peak at zero frequency scale differently with the THz field, which, if I understood correctly is important to differentiate the two components as arising from two different effects (or at least that there is some contribution from another effect which has a different scaling). I don't quite see why that has to be the case. In their response to Referee #2, they argue that if one normalizes the Fourier transform for each frequency component then the curvature should be the same. However, what the authors are describing seems different than the simple approach of scaling the time domain signal by the total field strength and then taking the Fourier transform. In this case, I would expect that different frequency components can certainly have different curvatures. Or am I missing something? I think this is a crucial point, so I would appreciate if the authors could explain the procedure and the argument in more detail.

4. Can the authors apply a band pass filter at a different frequency at fixed temperature to show that the magnetic effect goes away? This would be a relatively clean demonstration that the lattice contribution is the key aspect to the signal. This is a point that was also originally brought up by Referee #1 in their initial report. I understand that the temperature dependence tunes the phonon mode into and out of resonance, but as noted by the authors several other parameters also change when doing so (oscillator strength, nonlinear optical constants, etc.), some of which are only obtained from theory. I appreciate the challenge in finding the appropriate optics at THz frequencies and carrying out an extra set of measurements, but given the strength of the claims the authors make in this paper, if such measurements are possible, they should be carried out to verify the conclusions.

5. Clearly, a crucial aspect to interpreting the low frequency signal as coming partially from a dynamical multiferroic effect is ruling out the other parts of the signal that can be explained by χ^3 nonlinearity. I understand that the authors have another paper in review regarding this χ^3 effect and not wanting to pre-empt that publication. But, I think it is important to include at least a subset of the details of the theory/analysis of the modeling of this effect in the supplement here, for example, the symmetry of the χ^3 tensor as referenced in the main text. At the very least, I would urge that the figure they provide in the rebuttal showing the TKE, total response, and μ_{ph} to replace panels a and b of Figure 4, and to include accompanying text / equations

explaining how one obtains these curves.

6. The idea to normalize the $E_x \cdot E_y$ nonlinear optical response so as to match the 6 THz peak seems to me still not fully justified. I understand that normalizing in this way is convenient from the standpoint that you end up with unaccounted for signal at zero frequency which can be explained via the dynamical multiferroicity effect. However, I could have chosen to normalize the zero frequency peak instead and end up with too much signal at 6 THz. Again, I understand the motivation to do it the way the authors did, but it is not convincing. The most convincing evidence would be to calculate what the real Kerr response would be given the field strengths measured. It seems that the authors have been able to provide ab initio calculations of the nonlinear optical coefficients in the extended data. With this information and the measured values of the electric field strengths E_x and E_y , they should be able to obtain values of the χ^3 induced Kerr rotation in real angular units. This to me is quite critical to truly back up the claims in this paper.

7. The authors have not adequately addressed the gap between the theoretical and experimentally measured values of the magnetic moment, in my opinion. I agree that actually trying to measure, for example, the spin vs. orbital moment is beyond the scope of this work (and as yet, probably technically not feasible). However, I differ in opinion with the statement of the authors that providing a reasonable physical explanation is not required for this publication. The authors claim roughly a four order of magnitude difference between the state of the art theory and experiment. If it was a factor of two, I would agree that this can be discounted at the level of this paper, but this is a massive disconnect. In their rebuttal the authors describe the motion of the charged Ti ions as generating an orbital magnetism. While, in principle, this could generate an effective magnetic field or crystal field splitting which could impart an orbital moment to the system, I reiterate a statement I made in my first report, which is that STO is a d^0 compound, so there are no electrons to polarize by the orbital field. Orbital moments are well understood in transition metal compounds with orbitally degenerate ground states, but such moments would still require at least one electron to occupy the d manifold. This is, still to me one of the biggest holes in the understanding of this effect and why I still am not convinced the signal is of magnetic origin. If the authors can provide some plausible physical mechanism that could generate such a large moment, that would go a long way to pushing the paper towards publication.

A more minor point:

8. In their rebuttal and in the revised methods, the authors refer to Reference 46 to help explain the characterization of the polarization state via rotating the GaP crystal by 90 deg. As far as I can tell, this reference points to a paper which does not use electro-optic sampling in GaP. I think this should be corrected.

To summarize, I believe there are still some important issues related to the analysis / interpretation of the data that need to be resolved, and hence I cannot recommend publication.

Author Rebuttals to First Revision:

We thank Referees #1 and #2 for their very positive evaluation of our revised manuscript, and for their strong recommendation for publication in Nature. As requested by the Editor, we provide a point-by-point response to the issues raised by Referee #3.

Referee #3 (Remarks to the Author in blue, our response in black):

The authors have presented a revised manuscript with an extensive amount of extended data and supplementary information added. They have included more details about the pulse characterization methods. They have also added raw data, error bars, and new measurements of the probe polarization dependence, which are very helpful in understanding the signals they observe. I appreciate the authors thorough work, and I do think the revised manuscript is much improved. However, I still have a number of concerns regarding the analysis and interpretation of the data that would prevent me from recommended the paper for publication.

My main concerns are described below in no particular order.

We thank the referee for the positive feedback and for acknowledging the improvement of our manuscript.

1. As I stated in my initial report, the pump-probe penetration depth mismatch is a crucial factor in extracting the overall size of the magnetic moment that is extracted from these measurements. The authors argue that the probe wavelength is in the transparency region for STO, so that it penetrates the entire sample, while the pump penetration depth is on the order of a few microns at all measured temperatures. I'm not sure how the authors come to this conclusion regarding the probe penetration. Looking at measurements of the absorption coefficient near the band edge of STO [Gogoi et al., PRB 93, 075204 (2016)], the band gap at 300 K was measured to be 3.15 eV – quite close to the probe photon energy at 400 nm (3.1 eV). In addition, a strong absorption tail at lower energies was found. Extracting the absorption coefficient from these measurements seems to give a probe penetration depth of ~3 microns or so, roughly the same as the pump penetration. This contradicts the authors assumption that the probe penetration is on the order of millimeters and will scale the extracted magnetic moment accordingly.

We agree with the referee that the pump-probe penetration depth is crucial. However, in the paper cited by the referee, the STO sample considered there is clearly different from ours, and from the known literature values of the dielectric constant. Indeed, in the datasheet of the provider of the sample in the mentioned PRB paper <http://www.crystec.de/daten/srtio3.pdf>, it is specified that their STO products are “colorless to pale yellow”. Our sample is transparent with no shades of color. Hanzig et al. [J. Appl. Phys. 110, 064107 (2011)] showed that the indirect bandgap of as-grown bulk STO is reduced from 3.44 eV (360 nm) to 3.21 eV (386 nm) after 10 hours of vacuum annealing at 900 °C. We believe the STO used in the PRB might have been subjected to this kind of process, hence showing enhanced absorption at 400 nm with respect to our sample.

Below are two photographs of our probe beam going through air (“before STO”) and through the STO sample (“after STO”), as well as the photodiode signal measured on an oscilloscope. The transmitted intensity is approximately 60% of the incident one, consistent with a material with $n = 2.6$ and negligible imaginary part k [Ref. 57: Polyanskiy, M. N., "Refractive index

database," <https://refractiveindex.info>, "M. J. Dodge. Refractive Index, in M. J. Weber (ed.), Handbook of Laser Science and Technology, Volume IV, Optical Material: Part 2, CRC Press, Boca Raton, 1986 (as cited in Handbook of Optics, 3rd edition, Vol. 4. McGraw-Hill 2009)]. In fact, such a transmittance value at normal incidence can be obtained via transfer matrix method for a 500 μm -thick layer of STO for $n = 2.6 + i6.5 \cdot 10^{-6}$. In this case, the absorbed intensity results to be approx. 10%, corresponding to a penetration depth of 4.9 μm , much longer than the crystal thickness. To be more conservatives, an upper limit for the absorption can be obtained by considering the k_{max} value for which the penetration depth equals the crystal thickness, i.e. $k_{max} = 6.4 \cdot 10^{-5}$. Even in this worst-case scenario, approx. 24% of the incident intensity would be transmitted, which is clearly less than what we measured. Our transmittance results are also confirmed by the characterization provided by MTI Corporation, the company from which our sample was purchased, available at the following link: <https://www.mtixtl.com/images/products/detail/STOTcurve.png>

Therefore, we stand by our original claim and the objection of the referee, motivated by a reference which performed measurement on a different sample, does not apply to our work.

2. Related to the above observation that probe photon energy is, in fact, near the band edge, the value of the magneto-optical coefficient can be modified by the pump field. In transition metal oxides, magneto-optical transitions are often associated with vibronic transitions (especially at energies close to the band edge), which are strongly coupled with lattice excitations. Since the authors are directly driving a phonon with their pump, it is, to me, not unlikely at all that the magneto-optical coefficient may be altered as a result of this driving. I want to stress that this is not just a pedantic point – such effects are present and affect magneto-optical measurements even in well-established magnetic compounds. In this case, where the origin of magnetism is less direct, I think it is important that the authors are able to address this point and rule out such a scenario.

We understand the referee's concern; however, we stress again that with a 3.1 eV probe photon energy, we are far enough from the band edge in our sample (different from the one mentioned and discussed in point 1) to exclude such effects. To remove any doubt, we show here yet another set of measurements to rule out the possibility that such effects may be relevant in our case. In the figure below, we plot, similarly to what is done in the manuscript, the Kerr rotation in the frequency domain, comparing the cases of 400 nm and 800 nm probe wavelengths. The latter case corresponds to a photon energy of 1.55 eV, less than half of the direct bandgap. Besides an overall scaling factor due to the different value of the Verdet constant at the two wavelengths (already addressed in the previous rebuttal), the two curves have a very similar shape, excluding exotic behaviors due to the proximity of the probe energy to the bandgap.

3. The authors show that the peak at 6 THz and the peak at zero frequency scale differently with the THz field, which, if I understood correctly is important to differentiate the two components as arising from two different effects (or at least that there is some contribution from another effect which has a different scaling). I don't quite see why that has to be the case. In their response to Referee #2, they argue that if one normalizes the Fourier transform for each frequency component then the curvature should be the same. However, what the authors are describing seems different than the simple approach of scaling the time domain signal by the total field strength and then taking the Fourier transform. In this case, I would expect that different frequency components can certainly have different curvatures. Or am I missing something? I think this is a crucial point, so I would appreciate if the authors could explain the procedure and the argument in more detail.

We thank the referee for allowing us to clarify this issue, which we believe became overly complicated probably due to some phrasings in our previous rebuttal. We include here an additional figure to aid the discussion. At the risk of stating something obvious, we go step by step through the computation procedure. As probably implicit in the comment of the referee, and correctly, the curvature of the quadratic dependence of the response on the peak amplitude is directly related to the height of the peak in the frequency spectrum. Panel (a) in the above figure shows the spectrum of $E_x \cdot E_y$ obtained via EOS, so it is the spectrum of the expected $\chi(3)$ effect for a certain pump field amplitude (proportional to $E_x \cdot E_y$). The data is taken at the maximum terahertz field; after that, we computed the response for the lower field cases assuming the natural scaling: linear in the field, quadratic in the product of the two fields, without any additional scaling factor. By plotting the peak amplitudes at ω_- and ω_+ frequencies, we obtain the blue and, respectively, the red data in panel (b). Clearly, the two parabolas are almost identical, given that their amplitude in the frequency domain is basically the same.

The green and black curves in the same panel (b) are instead derived from the measurements of the Kerr rotation in STO. They are normalized with the same procedure described in the paper, i.e. by matching the data at high frequency peak ω_+ (black symbols). In this case, we actually only matched the data corresponding to the highest field data point, and then used the same scaling factor for the data measured at lower fields. Also, using the same and only

scaling factor, we normalized the low frequency peak data (green symbols). This procedure clearly shows the different curvature of the parabolas, and indicates that by considering the $\chi(3)$ response alone, one cannot retrieve the experimental trend. Hence, an additional mechanism must be present to explain the extra spectral weight at low frequencies.

To rephrase differently: we understood how the $\chi(3)$ scales, and we can compute it for each measurement we performed. In all cases, we note that, compared to the overall response, the $\chi(3)$ only response always shows a smaller amplitude in the low frequency peak, as if there is some extra spectral weight in the total response. In the field-dependent scans, we furthermore noticed that such extra spectral weight scales quadratically with the field. Hence, on top of the $\chi(3)$ response, there is an additional effect which goes as the square of the field. This is supported by the experimental data, it is not an interpretation. Then, we noted that such an effect is consistent with the prediction of dynamical multiferroicity, which is also a true statement. This, obviously, does not demonstrate uniquely that dynamical multiferroicity is at play, but it shows that it is consistent with that. If there was a different dependence (linear or cubic for instance, it would have indicated something inconsistent with dynamical multiferroicity. However, in the absence of any alternative mechanisms which neither we or the referee could suggest, which could be tested against ours, the simplest explanation (albeit far from simple) that we could find to explain our data is the one of dynamical multiferroicity.

4. Can the authors apply a band pass filter at a different frequency at fixed temperature to show that the magnetic effect goes away? This would be a relatively clean demonstration that the lattice contribution is the key aspect to the signal. This is a point that was also originally brought up by Referee #1 in their initial report. I understand that the temperature dependence tunes the phonon mode into and out of resonance, but as noted by the authors several other parameters also change when doing so (oscillator strength, nonlinear optical constants, etc.), some of which are only obtained from theory. I appreciate the challenge in finding the appropriate optics at THz frequencies and carrying out an extra set of measurements, but given the strength of the claims the authors make in this paper, if such measurements are possible, they should be carried out to verify the conclusions.

We thank the referee for the observation. In principle we could perform such a measurement, but it would be much time consuming, as we would need to basically re-build the optical setup. Going down to 1 THz, the DSTMS organic crystal used to generate the terahertz field that we used in the manuscript is not ideal, and one would also need to move to a different source. We definitely would have tried it, if it weren't for the fact that the optical parametric amplifier is currently not operating, and we cannot perform such a measurement.

However, we had previously already investigated a similar situation: the response of STO to a linearly polarized THz pump field centered at 1THz, 2THz and, 3THz, at room temperature. The results are reported in arXiv:2210.14053v2 and summarized in the figure below. When we go the furthest away from the phonon resonance (i.e. using a 1 THz center frequency for the pump), the THz field penetrates much deeper into the sample, and optical propagation effects due to pump-probe phase mismatch appear. The calculations of both $\chi(3)$ and dynamical multiferroicity responses in the presence of propagation effects is extremely complicated and beyond the scope of this work. There is enough novelty in the current manuscript, that it would be impossible to discuss that aspect as well. We stress though that the measurements shown below demonstrate that we can exclude any propagation effects,

further proving the point of total mismatch between penetration depth of pump and probe discussed above.

Finally, from the mathematical description of a forced oscillator, if the material properties are known with enough accuracy, there is no reason to expect a qualitatively different behavior whether the resonance is hit moving the driving field center frequency, or moving the oscillator center frequency. We demonstrated in our manuscript and in the rebuttal letters that we were able to carefully characterize the relevant optical properties of our sample in detail. Hence, there is no reason to expect a different behavior from the point of view of the oscillator response.

{REDACTED}

5. Clearly, a crucial aspect to interpreting the low frequency signal as coming partially from a dynamical multiferroic effect is ruling out the other parts of the signal that can be explained by chi3 nonlinearity. I understand that the authors have another paper in review regarding this chi3 effect and not wanting to pre-empt that publication. But, I think it is important to include at least a subset of the details of the theory/analysis of the modeling of this effect in the supplement here, for example, the symmetry of the chi3 tensor as referenced in the main text. At the very least, I would urge that the figure they provide in the rebuttal showing the TKE, total response, and mu_ph to replace panels a and b of Figure 4, and to include accompanying text / equations explaining how one obtains these curves.

In the already mentioned manuscript M. Basini et al. [arXiv:2210.14053v2], some of us showed that the electronic Kerr effect (EKE) response is given by

$$\Delta\Gamma^e \sim \frac{1}{4} [E_x^2 - E_y^2] \Delta\chi \sin(4\vartheta) + 2E_x E_y \left[\chi^{(3)}_{ijj} + \frac{1}{2} \Delta\chi \sin^2(2\vartheta) \right],$$

where E_x , E_y are the components of the pump pulse along generic x and y orthogonal directions, ϑ is the angle that x and y form with respect to the main crystallographic axes (i and j) and $\Delta\chi \equiv \chi^{(3)}_{iii} - 3\chi^{(3)}_{ijj}$, being $\chi^{(3)}_{ijj} \simeq 0.47\chi^{(3)}_{iii}$ the only two independent tensor

components of the $\chi^{(3)}$ tensor in cubic STO [M. J. Weber, *Handbook of Optical Materials* (CRC Press, 2002)]. If $\vartheta = 45^\circ$, one finds that $\Delta\Gamma^e \sim 2E_x E_y \left[\chi^{(3)}_{iijj} + \frac{1}{2}\Delta\chi \right]$, and the signal is proportional to the product of the THz pump field components along perpendicular directions. In case of circularly polarized light of opposite helicity (i.e. LCP and RCP), the signal difference $\Delta\Gamma^e(LCP) - \Delta\Gamma^e(RCP)$ is still proportional to $E_x E_y$, since only one of the two pump components changes sign.

Besides the EKE, it has been shown that an additional contribution, named ionic Kerr effect (IKE), associated with the nonlinear excitation of the IR-active soft phonon mode, is present. The IKE response $\Delta\Gamma^{ph}$ can be effectively modeled by replacing the E_x, E_y components in $\Delta\Gamma^e$ with a convolution between the pump and the single or two-phonon propagators, accounting for the intermediate second-order excitation of the soft mode. Moreover, the $\chi^{(3)}$ tensor should be replaced with an effective nonlinear coupling between the pump and probe pulses and the IR-active phonon. While the time evolution of the IKE can be well reproduced once the time profile of the pump pulse and the spectral features of the soft phonon mode are known, in order to estimate the relative weight of the IKE and the EKE in a rigorous way, an *ab initio* estimation of the effective nonlinear coupling is needed. This would require a state of art extension of the available DFT codes, that has been investigated only recently (see e.g. PRB 107, 174307 (2023)) and goes far beyond the scope of this work. For this reason, we decided to model the full Kerr response with the electronic contribution only, provided that the ionic contribution has a similar spectral content, as not to introduce any free adjustable parameter in our simulations.

For completeness, and to follow the referee suggestion, we decided to show the full Kerr effect, including both the EKE and the IKE, in the Methods and the Extended Data, even if our original plan was to discuss these results in a related work [arXiv:2210.14053v2].

6. The idea to normalize the $E_x E_y$ nonlinear optical response so as to match the 6 THz peak seems to me still not fully justified. I understand that normalizing in this way is convenient from the standpoint that you end up with unaccounted for signal at zero frequency which can be explained via the dynamical multiferroicity effect. However, I could have chosen to normalize the zero frequency peak instead and end up with too much signal at 6 THz. Again, I understand the motivation to do it the way the authors did, but it is not convincing. The most convincing evidence would be to calculate what the real Kerr response would be given the field strengths measured. It seems that the authors have been able to provide *ab initio* calculations of the nonlinear optical coefficients in the extended data. With this information and the measured values of the electric field strengths E_x and E_y , they should be able to obtain values of the χ^3 induced Kerr rotation in real angular units. This to me is quite critical to truly back up the claims in this paper.

We agree with the referee on the general importance of having quantitative information on the χ^3 -induced Kerr rotation in angular units. However, we point out that we don't have any *ab initio* calculation for the value of the nonlinear optical coefficient. We also stress that, for our study, such information is not crucial, as we do not expect any magnetic contribution with spectral components at around 6 THz.

Regarding the choice of our normalization compared to the other scenario described by the referee, i.e. by choosing to normalize on the low-frequency peak. If we were to normalize on the low-frequency peak, we would obviously obtain the opposite trend. Namely, at the temperature for which the pump is in resonance with the phonon, the amplitude of the 6 THz peak would be lower than the one expected from a $\chi(3)$ response. We cannot see how this would help, and it would lack a physical interpretation which can instead be given with our original choice. In fact, with that choice, we are consistent with the facts that: (i) we expect no contribution due to the phonon magnetic moment at 6 THz, and (ii) we are able to reproduce the temperature dependence of the 6 THz component by modeling the nonlinear optical response of STO in our experimental configuration, i.e. TKE+IKE (see arXiv:2210.14053v2 and answer to question 5).

7. The authors have not adequately addressed the gap between the theoretical and experimentally measured values of the magnetic moment, in my opinion. I agree that actually trying to measure, for example, the spin vs. orbital moment is beyond the scope of this work (and as yet, probably technically not feasible). However, I differ in opinion with the statement of the authors that providing a reasonable physical explanation is not required for this publication. The authors claim roughly a four order of magnitude difference between the state of the art theory and experiment. If it was a factor of two, I would agree that this can be discounted at the level of this paper, but this is a massive disconnect. In their rebuttal the authors describe the motion of the charged Ti ions as generating an orbital magnetism. While, in principle, this could generate an effective magnetic field or crystal field splitting which could impart an orbital moment to the system, I reiterate a statement I made in my first report, which is that STO is a d0 compound, so there are no electrons to polarize by the orbital field. Orbital moments are well understood in transition metal compounds with orbitally degenerate ground states, but such moments would still require at least one electron to occupy the d manifold. This is, still to me one of the biggest holes in the understanding of this effect and why I still am not convinced the signal is of magnetic origin. If the authors can provide some plausible physical mechanism that could generate such a large moment, that would go a long way to pushing the paper towards publication.

There are two main points to this issue, and we break it down into smaller parts to address it properly. First, the four orders of magnitude discrepancy between measured and simulated magnetic moment. Second, the polarization of the electrons in a d0 compound.

As to the first point, the four orders of magnitude difference is implicitly present already in the 1905 paper by Barnett. The recent arXiv manuscript [C.S. Davies et al, Phononic Switching of Magnetization by the Ultrafast Barnett Effect, arXiv:2305.11551] mentioned in our previous rebuttal where a ferromagnet is switched via the same effect that we report here, has been completely ignored by the referee and the objection reiterated as that work did not appear in the meanwhile. We state it again: if a transfer of mechanical angular momentum (using the same notation of Fig. 4) $\mathbf{L} = \mathbf{L}_{el} + \mathbf{L}_{ph}$ happens between phonons and electrons, the lighter electronic mass causes the boosting of the magnetic moment $\mathbf{M}_{el} = \gamma_{el} \mathbf{L}_{el}$ by roughly the ratio of nucleus to electron masses, which is about a factor 4000. In other words, the transfer of angular momentum from the ions to the electrons causes a very large enhancement of \mathbf{M}_{el} , because the gyromagnetic ratio γ_{el} is much higher for electrons than for ions. This fact has been actually discussed already in two theoretical papers: in one from Rebane [T. Rebane, Faraday effect produced in the residual ray region by the magnetic moment of an optical

phonon in an ionic crystal, *Zh. Eksp. Teor. Fiz.* **84**, 2323 (1983)], and in one more closely related to the current work, from Geilhufe et al. [Dynamically induced magnetism in KTaO_3 , *Physical Review Research* **3**, L022011 (2021)], Ref. 34 of the last version of the manuscript. That work, in which two of us are co-authors, considered another perovskite with a similar soft phonon mode, namely KTaO_3 . The only assumption made to describe the coupled electron-nucleus system was that the center of mass is not affected by the photon field and is conserved during the circular motion. With this assumption, the expected enhancement due to the “phonon” Barnett effect is given in Eq. (13) in Ref. 34, and it is predicted to be between three and five orders of magnitude. The general mechanism presented there is also valid for this work, where for the first time we observe it experimentally and with much greater detail. Hence, we completely disagree with the statement that we did not provide a reasonable physical explanation: it is there and in the references that we cited.

To the second point. Based on the mechanism above, an orbital electronic motion is expected to give rise to an orbital electronic moment. This orbital motion and magnetic moment of “current loop” would produce the magnetic field. The effect would be present for all electrons on the unit cell and does not require the d states to be occupied. Even though we do not want to speculate too much about this, the orbital motion of any electron can induce the Kerr effect. Now the question from the referee is: is the effect that we observe due to spin polarized electrons? As just stated, the orbital origin of the effect does not require spin polarization of the d orbitals. In addition, we reiterate that we are not in an equilibrium situation, and that the distortion of the electronic cloud due to the phonon nonlinearities could change the equilibrium state. Particularly in the presence of spin-orbit coupling which could be large on the heavy Sr ions. (We note that the picture of only the Ti ion moving is a simplified one: obviously the soft phonon mode involves the motion of the atoms in the entire unit cell, in particular it is the movement of the oxygen cage with respect to the Ti and Sr ones.) We do not have a clear picture of how this would happen, but neither does the referee provide an alternative picture for the effect that we see. We also think it is fair to leave the spin vs orbit discussion to a later work, as the referee also acknowledges.

We would also like to point out that we agree with the referee that there are still question marks left. But after having turned over all the stones, and with compelling evidence from independent works showing a picture consistent with ours, either we leave the paper in a drawer, or we put it out acknowledging that still much has to be understood. This is after all, the first observation of the effect. It would be even more interesting if this observation will open new possibilities in nonlinear phononics. Even if we were to be missing some aspects with the interpretation, there is the fact that a thin film ferromagnet has been switched driving circular motions, with the approach that we discovered. The details may be different, but the effect is there.

A more minor point:

8. In their rebuttal and in the revised methods, the authors refer to Reference 46 to help explain the characterization of the polarization state via rotating the GaP crystal by 90 deg. As far as I can tell, this reference points to a paper which does not use electro-optic sampling in GaP. I think this should be corrected.

We thank the referee for noticing this typo and we corrected it. This is indeed ref. 51, not 46.

To summarize, I believe there are still some important issues related to the analysis / interpretation of the data that need to be resolved, and hence I cannot recommend publication.

We thank the referee for the many comments. We believe, together with the other two referees, that the paper was already suitable for publication at the previous round, but we appreciate the request for clarification that may have helped improve the form of the paper even further. The main claims have however remained the same, and as time passes by, several other groups are independently reporting effects similar to the one we report here, as pointed out by one of the other referees as well.

We believe that the referee is trying to get answers to his or her questions. In the meantime, there is a growing body of literature coming in support of the observed effect. We are perplexed by the fact that in this rebuttal, the referee seems to have completely ignored the results of the recent arXiv paper which we indicated in our previous rebuttal and which demonstrates that a thin film ferromagnet can switch its magnetization direction driving circular phonons on the substrate. At the very least the fact that phonons can drive a magnetic state is now proven in more than one setting. This observation is new, and we believe that the wider community deserves to be informed. As stated above, we acknowledge that much more is to be understood. However, we are of the opinion that the recent experimental evidence, closely related to our work and consistent with our claim, should not be dismissed and ignored by the referee. It does provide an independent confirmation the effect is real, large and possibly applicable to a wide class of materials.

Finally, we also stress that in two reports, the referee has not provided an alternative explanation for the effect we see, and which we could have considered and tested against others. Instead, several potential errors in our work were pointed out by the referee and we demonstrated, one after one, that there were no errors. Finding errors is part of the job of a referee, and we appreciate that. However, after two extremely lengthy rebuttals with multiple additional measurements, if there are no alternative hypotheses to be discussed, it is reasonable that the only one which is on the table should be accepted.

Reviewer Reports on the Second Revision:

Referees' comments:

Referee #2:

supportive of publication

Referee #3 (Remarks to the Author):

I thank the authors for taking the time to review and respond to my comments. They have frankly done an outstanding job of clarifying some subtle aspects of the work through very useful explanations and data (both in this rebuttal and the previous one) that have helped to answer some of the questions I had.

For example, I can see now that my concerns regarding the probe wavelength and the penetration depth/magneto-optical effects can be disregarded. I appreciate the clarification regarding potentially different sample parameters from the work I cited. In this context, their measurement of the rotation signal using 800 nm probe matching the 400 nm probe is a highly relevant result. I think this data should be included in the supplement, if possible.

I am less convinced by the scaling argument the authors present and still have my doubts that the low frequency excess spectral weight that the authors attribute to a magnetic moment could not arise from a frequency-dependent nonlinear contribution. Let me explain.

First, the way I understand what the authors did is the following: the authors measure $E_x(t)$ and $E_y(t)$ by means of electro-optic sampling, from which they can compute the spectrum of the product $E_x E_y$. The authors then extract the relative amplitudes of the peaks at ω_+ (6 THz) and ω_- (~ 0 THz). They assume that the spectral amplitudes of at each of these frequencies will scale linearly with E_x and E_y (hence a quadratic scaling with $E_x E_y$), from which they can get the expected peak amplitudes for ω_+ and ω_- for different incident THz field strengths. Already here, while of course one should expect that these amplitudes will scale as the authors have assumed, I'm a bit puzzled as to why the authors did not just measure the EOS traces with different incident THz fields, so they could get the scaling directly. This would help rule out any potential artifacts in the EOS. I don't think this would have been too hard a measurement, but I could be wrong. I guess one needs to assume here that the spectrum looks identical if I choose to use either $E_x E_y$ taken from the RCP THz pump or the LCP pump. Is that the case?

Continuing, the authors measure the polarization rotation in STO induced by the left and right circularly polarized pumps and determine the signal arising from the difference, $S(t)$. The authors then get the spectrum of S , choosing to normalize this spectrum such that the height of the ω_+ peak of S matches that of $E_x E_y$ at one field strength. They then see how the ω_+ and ω_- peaks of S scale in comparison to those of $E_x E_y$. They find that the ω_- peak of S has extra spectral weight than the corresponding peak in $E_x E_y$, which also scales quadratically with incident field.

Importantly, based on the analysis in [arXiv:2210.14053v2], the authors show that the "trivial" nonlinear optical response (coming from the Kerr effect) should scale like $\chi^3 E_x E_y$ in the signal channel. Hence, they attribute the extra spectral weight to a different effect, namely an induced magnetic moment arising from the "dynamical multiferroicity" effect. (I think any confusing regarding this analysis could be made explicit by replacing Figure 4a and b with Extended Figure 8a and b).

Ok, assuming I understood the experiment and analysis properly, I am really sorry to say that I still think such an exotic explanation is not fully justified. A potentially much simpler explanation is that there is a frequency dependence to χ^3 (and/or the ionic part of the Kerr effect). The dielectric function of STO is highly frequency dependent in the region of the measurement, as shown in Extended Data Fig. 6. For example, the real part is large and positive at the ω_- frequency

while it is negative close to ω . I understood that the authors do not have a way to calculate the nonlinear optical coefficients, but based on simple arguments, one can expect that χ_3 will be related to the linear susceptibility (i.e. $\chi_1 = \epsilon - 1$) at the detection frequency. So, shouldn't we expect that from the Kerr effect alone, the polarization dependence will have a non-trivial frequency dependence that differs from simply multiply $E_x E_y$ by a constant factor?

This point is related to ones I brought up previously. Ultimately, in order to be convinced that this is not a nonlinear optical effect, in my opinion, the authors need to be able to calculate the "trivial" χ_3 response quantitatively, so they can subtract it off from their signal to reveal the true unaccounted for signal, that could then be attributed to more exotic effects. I think all of us who work in the field of ultrafast/nonlinear THz spectroscopy are used to seeing signals that have strange spectral shapes or features. On closer inspection, it is often found that these are a result of the strong dispersion/absorption in the THz, propagation effects, frequency-dependent nonlinear contributions, etc. I've brought these points up in previous reports, and I understand that they are difficult to fully account for, but again, unless such more mundane issues can be truly ruled out, I have a hard time buying into the proposed magnetic mechanism. Let's forget about propagation effects for now as the authors stated this is beyond the scope of the work. Can a frequency-dependent χ_3 (or more generally, a frequency-dependent purely optical nonlinear response) be ruled out?

Look, I understand that this must be frustrating for the authors. If one accepts the analysis, the dynamical multiferroic explanation they put forward seems generally consistent with the data. However, there are still other more trivial explanations. Having looked at this paper and all the authors rebuttals and revisions from many angles, I am still left with my reservations. I accept there should be some point where the battle is won, so I think at this stage I would be satisfied if the authors can help me understand how to discount the possibility of a frequency-dependent χ_3 response being responsible for the extra low frequency signals.

I am also well aware of the arxiv paper of Davies et al. I think it is (for lack of a better word) rather dodgy to use an as yet peer-reviewed work to provide justification for this one, so I have chosen not to take it into consideration in this review. But, I will say that, as far as I can tell, one of the key differences is that the Davies work provides direct evidence of a magnetic effect, which can be seen through the switching of a ferromagnetic layer, without relying on an optical readout which could be subject to the nonlinear optical issues discussed above. In that sense, the argument is stronger for the proposed dynamical multiferroic mechanism. I am not implying that the authors of the current paper should try to replicate that experiment, I am just saying that each paper needs to be evaluated and scrutinized under its own merit.

I hope my current objections and considerations are clear. To summarize, despite the additional explanations and data provided in the most recent rebuttal and revision, I still would be uncomfortable recommending publication.

Author Rebuttals to Second Revision:

Rebuttal to Referee #3

I thank the authors for taking the time to review and respond to my comments. They have frankly done an outstanding job of clarifying some subtle aspects of the work through very useful explanations and data (both in this rebuttal and the previous one) that have helped to answer some of the questions I had.

We thank the referee for acknowledging our work in considering her or his opinion. We believe we have now addressed all of his or her concerns, and we hope he or she will recommend the publication of our revised manuscript.

For example, I can see now that my concerns regarding the probe wavelength and the penetration depth/magneto-optical effects can be disregarded. I appreciate the clarification regarding potentially different sample parameters from the work I cited. In this context, their measurement of the rotation signal using 800 nm probe matching the 400 nm probe is a highly relevant result. I think this data should be included in the supplement, if possible.

We thank the Referee for admitting that his or her concerns can be disregarded. Following the suggestion, we have included the data taken with the 800 nm probe in the Extended Data.

I am less convinced by the scaling argument the authors present and still have my doubts that the low frequency excess spectral weight that the authors attribute to a magnetic moment could not arise from a frequency-dependent nonlinear contribution. Let me explain.

First, the way I understand what the authors did is the following: the authors measure $E_x(t)$ and $E_y(t)$ by means of electro-optic sampling, from which they can compute the spectrum of the product $E_x \cdot E_y$. The authors then extract the relative amplitudes of the peaks at ω_+ (6 THz) and ω_- (~0 THz). They assume that the spectral amplitudes of at each of these frequencies will scale linearly with E_x and E_y (hence a quadratic scaling with $E_x \cdot E_y$), from which they can get the expected peak amplitudes for ω_+ and ω_- for different incident THz field strengths. Already here, while of course one should expect that these amplitudes will scale as the authors have assumed, I'm a bit puzzled as to why the authors did not just measure the EOS traces with different incident THz fields, so they could get the scaling directly. This would help rule out any potential artifacts in the EOS. I don't think this would have been too hard a measurement, but I could be wrong. I guess one needs to assume here that the spectrum looks identical if I choose to use either $E_x \cdot E_y$ taken from the RCP THz pump or the LCP pump. Is that the case?

We acknowledge the question from the Referee. However, there are no artifacts in the EOS, and we check the EOS sampling routinely in our lab to make sure the crystals are not damaged. Since the lab started operating, in 2017, we had to replace only one GaP crystal and because we damaged it by mistake with a too intense probe beam. Those are routine measurements that have been well known to the terahertz community for decades. We plot them here below for the case of a broadband field exceeding the MV/cm (larger than the largest field used in the manuscript).

Normalized EOS signal as a function of the relative angle between two polarizers

EOS signal divided by the value of the cosine square, since the rotating polarizer was between two fixed polarizers in the setup. The almost perfect overlap (the error has to be ascribed to non-perfect orientation of the polarizers) demonstrates the linearity of the EOS up to the maximum field (for this trace exceeding 1 MV/cm).

Continuing, the authors measure the polarization rotation in STO induced by the left and right circularly polarized pumps and determine the signal arising from the difference, $S(t)$. The authors then get the spectrum of S , choosing to normalize this spectrum such that the height of

the w^+ peak of S matches that of $E_x E_y$ at one field strength. They then see how the w^+ and w^- peaks of S scale in comparison to those of $E_x E_y$. They find that the w^- peak of S has extra spectral weight than the corresponding peak in $E_x E_y$, which also scales quadratically with incident field.

Importantly, based on the analysis in [arXiv:2210.14053v2], the authors show that the “trivial” nonlinear optical response (coming from the Kerr effect) should scale like $\chi^3 E_x E_y$ in the signal channel. Hence, they attribute the extra spectral weight to a different effect, namely an induced magnetic moment arising from the “dynamical multiferroicity” effect. (I think any confusing regarding this analysis could be made explicit by replacing Figure 4a and b with Extended Figure 8a and b).

Ok, assuming I understood the experiment and analysis properly, I am really sorry to say that I still think such an exotic explanation is not fully justified. A potentially much simpler explanation is that there is a frequency dependence to χ^3 (and/or the ionic part of the Kerr effect). The dielectric function of STO is highly frequency dependent in the region of the measurement, as shown in Extended Data Fig. 6. For example, the real part is large and positive at the w^- frequency while it is negative close to w^+ . I understood that the authors do not have a way to calculate the nonlinear optical coefficients, but based on simple arguments, one can expect that χ^3 will be related the linear susceptibility (i.e. $\chi^1 = \epsilon - 1$) at the detection frequency. So, shouldn't we expect that from the Kerr effect alone, the polarization dependence will have a non-trivial frequency dependence that differs from simply multiply $E_x E_y$ by a constant factor?

This point is related to ones I brought up previously. Ultimately, in order to be convinced that this is not a nonlinear optical effect, in my opinion, the authors need to be able to calculate the “trivial” χ^3 response quantitatively, so they can subtract it off from their signal to reveal the true unaccounted for signal, that could then be attributed to more exotic effects. I think all of us who work in the field of ultrafast/nonlinear THz spectroscopy are used to seeing signals that have strange spectral shapes or features. On closer inspection, it is often found that these are a result of the strong dispersion/absorption in the THz, propagation effects, frequency-dependent nonlinear contributions, etc. I've brought these points up in previous reports, and I understand that they are difficult to fully account for, but again, unless such more mundane issues can be truly ruled out, I have a hard time buying into the proposed magnetic mechanism. Let's forget about propagation effects for now as the authors stated this is beyond the scope of the work. Can a frequency-dependent χ^3 (or more generally, a frequency-dependent purely optical nonlinear response) be ruled out?

We thank the referee for summarizing the essence of the problem, and for forcing us to think more thoroughly on how to rule out possible artifacts. We first reiterate something that was put forward since the very first version of this manuscript: the temperature-dependent data. By moving the phonon response with temperature, we strongly changed the frequency-dependence of epsilon, and the mechanism we propose is very robust against such changes. As we show in Fig. 4c, the “extra weight” in the low-frequency peak of the spectrum follows the same trend as the soft-phonon frequency. If it was the value of epsilon at low-frequency that mattered, then the lower the temperature, the larger the low-frequency response should have been observed. The experimental evidence seems to directly contradict the suggestion by the Referee.

Nevertheless, we believe that we have found an additional way to rule out a frequency-dependent $\chi(3)$, only based on symmetry and on the experimental data. In the related manuscript M. Basini et al. [arXiv:2210.14053v2], where we discuss the ionic and electronic Kerr effects (now mentioned in the additional data of this manuscript as well), we used a linearly polarized terahertz electric field to fully map the response of STO by rotating the sample in plane. The agreement between the model that we developed and the experiments is outstanding, demonstrating that we fully understand the $\chi(3)$ tensor and its symmetry, and that our simulations catch well the dynamics except for one single scaling factor.

We now can use this fact to calculate the “trivial” $\chi(3)$ response by calculating the expected response of a linearly polarized light, where dynamical multiferroicity is not active. Then, use the same formalism and the same scaling factor to calculate the response for circularly-polarized light, and compare it to experiments. If there is a mismatch between the two, then any additional features cannot be related to an arbitrary frequency-dependence of $\chi(3)$, which is a uniquely defined tensor describing the material, and must be attributed to additional effects.

This is shown in the two panels below, where the simulation data is scaled so that the two responses for linearly polarized light have the same peak amplitudes. The input terahertz fields for the simulations are the ones retrieved from the EO sampling.

From the data alone, it is already clear that the difference of the two circular polarizations (RCP - LCP)/2 has a larger amplitude in the experimental data than in the simulations. To appreciate this better, we subtract the experimental and simulated curves, and obtain the figure below.

The data clearly shows that there are two extra “bumps”, one positive and one negative. The first one, positive, as discussed in a previous rebuttal is associated with the ionic Kerr effect, discussed in M. Basini et al. [arXiv:2210.14053v2], similar to what is observed also in the linear case. The second is what we claim to be the dynamical multiferroicity contribution. We could in principle simulate the ionic Kerr effect, and include even the first bump in the simulations, but we prefer to preserve simplicity than adding other parameters to obtain a better figure. We have performed this simulation in the previous rebuttal.

We note that the data is not the same as the one in the manuscript: we looked for earlier measurements where we had performed experiments with circularly and linearly polarized light one after another, so as to minimize drift and have a reliable comparison between the different runs.

Look, I understand that this must be frustrating for the authors. If one accepts the analysis, the dynamical multiferroic explanation they put forward seems generally consistent with the data. However, there are still other more trivial explanations. Having looked at this paper and all the authors rebuttals and revisions from many angles, I am still left with my reservations. I accept there should be some point where the battle is won, so I think at this stage I would be satisfied if the authors can help me understand how to discount the possibility of a frequency-dependent χ_3 response being responsible for the extra low frequency signals.

We hope that our explanation provided earlier suffices for the Referee. We appreciate the Referee's willingness to avoid publishing a paper that contains obvious mistakes. However, after the extensive rebuttals and the amount of effort we put into it, we hope that the Referee will acknowledge our efforts to examine every possibility. We spent two years attempting to identify

all of the "artifacts" we observed, which the community had overlooked until now. We are also not attempting to promote an unconventional story; dynamical multiferroicity must exist, and the conservation of angular momentum must be maintained. We believe that we have proposed the simplest (although not simple) explanation that we were able to find, and a large part of the community has acknowledged our findings even before the paper was published. The paper has already received ten citations while being on the arXiv, and a new manuscript by Roberto Merlin (<https://arxiv.org/abs/2309.13622>) proposes an even more exotic explanation involving "Non-Maxwellian Magnetic-esque Fields." The suggestion is intriguing, and we cite this new manuscript in the revised version of our paper as an alternative explanation. However, we feel that at present, we cannot refute either hypothesis, and we maintain our original explanation as the primary one.

I am also well aware of the arxiv paper of Davies et al. I think it is (for lack of a better word) rather dodgy to use an as yet peer-reviewed work to provide justification for this one, so I have chosen not to take it into consideration in this review. But, I will say that, as far as I can tell, one of the key differences is that the Davies work provides direct evidence of a magnetic effect, which can be seen through the switching of a ferromagnetic layer, without relying on an optical readout which could be subject to the nonlinear optical issues discussed above. In that sense, the argument is stronger for the proposed dynamical multiferroic mechanism. I am not implying that the authors of the current paper should try to replicate that experiment, I am just saying that each paper needs to be evaluated and scrutinized under its own merit.

Although we acknowledge that every paper has its own value, we cannot ignore the striking similarity between their findings and our own results. Both studies involve the pumping of circular phonons in the substrate, which leads to enhancement. We fail to understand why the interpretation made in their study (which claims the same Barnett effect) is considered valid, while ours is not. Additionally, the data obtained from the ferromagnetic material, which acts as a magnetic field sensor, suggests that the magnetic fields generated are real, contradicting the idea that they are non-Maxwellian. However, we must admit that this is still too speculative at the moment.

I hope my current objections and considerations are clear. To summarize, despite the additional explanations and data provided in the most recent rebuttal and revision, I still would be uncomfortable recommending publication.

We believe that we have addressed all the doubts expressed by the Referee. We hope that the Referee is now convinced that the data is consistent with our interpretation and will agree on recommending publication of our manuscript.

Reviewer Reports on the Third Revision:

Referees' comments:

Referee #3 (Remarks to the Author):

The authors have been thorough in their responses to my questions, and although I have a few lingering doubts, I would be fine with the paper moving towards publication at this stage. I appreciate the authors' patience throughout this process and taking the time and effort to carefully consider my comments and critiques at every point.

There were a number of important notes of exposition and pieces of data that the authors presented during the review process. I believe a subset of these are critical to providing a full and complete picture of the experiments and interpretation, and I would therefore strongly recommend that they be included in the final manuscripts, if possible:

- As stated in my previous report, I think current Figure 4a,b in the main text should be replaced by the current Extended Fig. 9a,b. The figures in the extended data show much more clearly what the contributions to the signal arise from, especially since the current Fig. 4b has the same information as Fig. 2b.
- In the authors' previous response, they showed the "Kerr effect" signals as a function of fluence (from Fig. 3b), compared with the fluence dependence of the same frequency components from the "linear" field measurement of $E_x \cdot E_y$. This data, I think, makes it much more clear why the scaling of the w^- frequency component is anomalous. The scaling shown in Fig. 3b on its own I think is a bit ambiguous, so I would strongly recommend that the authors overlay the fluence dependence of the w^- and w^+ signals from $E_x \cdot E_y$ on Fig. 3b as in the panel b of the figure on page 4 of the previous rebuttal.
- The new data/simulations the authors provide on pages 4 and 5 of the current rebuttal do a good job of "ruling out" a χ^3 response and should be included in the extended data/supplement.

Author Rebuttals to Third Revision:

This document contains our point-by-point responses (**in black**) to Referee 3's comments (**in blue**).

Referee #3 (Remarks to the Author):

The authors have been thorough in their responses to my questions, and although I have a few lingering doubts, I would be fine with the paper moving towards publication at this stage. I appreciate the authors' patience throughout this process and taking the time and effort to carefully consider my comments and critiques at every point.

We thank the Referee for his positive evaluation of our work and supporting its publication.

There were a number of important notes of exposition and pieces of data that the authors presented during the review process. I believe a subset of these are critical to providing a full and complete picture of the experiments and interpretation, and I would therefore strongly recommend that they be included in the final manuscripts, if possible:

1) As stated in my previous report, I think the current Figure 4a,b in the main text should be replaced by the current Extended Fig. 9a,b. The figures in the extended data show much more clearly what the contributions to the signal arise from, especially since the current Fig. 4b has the same information as Fig. 2b.

We agree with the Referee and we implemented the changes as suggested. The main text has now slightly changed to describe the new figure correctly.

2) In the authors' previous response, they showed the "Kerr effect" signals as a function of fluence (from Fig. 3b), compared with the fluence dependence of the same frequency components from the "linear" field measurement of $E_x E_y$. This data, I think, makes it much more clear why the scaling of the ω -frequency component is anomalous. The scaling shown in Fig. 3b on its own I think is a bit ambiguous, so I would strongly recommend that the authors overlay the fluence dependence of the ω - and ω + signals from $E_x E_y$ on Fig. 3b as in the panel b of the figure on page 4 of the previous rebuttal.

We followed the Referee's suggestion and made the requested changes in both the figure and in the main text to reflect those changes.

3) The new data/simulations the authors provide on pages 4 and 5 of the current rebuttal do a good job of "ruling out" a χ^3 response and should be included in the extended data/supplement.

We agree with the Referee and we added this information as a new Extended Data Figure 7 (in the new numbering that we adopted to follow the Editorial guidelines), and described it in the Methods.